# Clinical use of an emergency manual by resuscitation teams and impact on performance in the emergency department: a prospective mixed-methods study protocol

Eric Dryver ,[1,2,3] Pontus Olsson de Capretz,[1,2] Mohammed Mohammad,[1] Malin Armelin,[1] William D. Dupont,[4] Anders Bergenfelz,[2] Ulf Ekelund[1,2]

[1]Department of Emergency and Internal Medicine, Skåne University Hospital Lund, Lund, Sweden
[2]Department of Clinical Sciences at Lund, Lund University, Lund, Sweden
[3]Practicum Clinical Skills Centre, Lund, Sweden
[4]Department of Biostatistics, Vanderbilt University Medical Center, Nashville, Tennessee, USA

**Correspondence to**
Dr Eric Dryver;
Eric.dryver@med.lu.se

## ABSTRACT

**Introduction** Simulation-based studies indicate that crisis checklist use improves management of patients with critical conditions in the emergency department (ED). An interview-based study suggests that use of an emergency manual (EM)—a collection of crisis checklists—improves management of clinical perioperative crises. There is a need for in-depth prospective studies of EM use during clinical practice, evaluating when and how EMs are used and impact on patient management.

**Methods and analysis** This 6-month long study prospectively evaluates a digital EM during management of priority 1 patients in the Skåne University Hospital at Lund's ED. Resuscitation teams are encouraged to use the EM after a management plan has been derived ('Do-Confirm'). The documenting nurse activates and reads from the EM, and checklists are displayed on a large screen visible to all team members. Whether the EM is activated, and which sections are displayed, are automatically recorded. Interventions performed thanks to Do-Confirm EM use are registered by the nurse. Fifty cases featuring such interventions are reviewed by specialists in emergency medicine blinded to whether the interventions were performed prior to or after EM use. All interventions are graded as indicated, of neutral relevance or not indicated. The primary outcome measures are the proportions of interventions performed thanks to Do-Confirm EM use graded as indicated, of neutral relevance, and not indicated. A secondary outcome measure is the team's subjective evaluation of the EM's value on a Likert scale of 1–6. Team members can report events related to EM use, and information from these events is extracted through structured interviews.

**Ethics and dissemination** The study is approved by the Swedish Ethical Review Authority (Dnr 2022-01896-01). Results will be published in a peer-reviewed journal and abstracts submitted to national and international conferences to disseminate our findings.

**Trial registration number** NCT05649891.

## INTRODUCTION
### Checklists
A checklist is a cognitive aid—consisting of a list of assessments or actions—designed

## STRENGTHS AND LIMITATIONS OF THIS STUDY

⇒ To date, the majority of studies of emergency manuals (EMs)—collections of crisis checklists— have been simulation-based, with the notable exceptions of one interview-based and one prospective study. This study prospectively evaluates an EM during the management of unselected, consecutive priority 1 patients in an emergency department over a 6-month period.

⇒ Some of the data on patient characteristics and EM use are automatically registered, while other EM use data are actively registered by the documenting nurse.

⇒ The study aims to determine whether interventions carried out thanks to checklist use are indicated, of neutral relevance or not indicated based on the clinical context, as assessed by external reviewers.

⇒ The external review process is based on information available from patient records, which may not convey all the relevant factors affecting clinical decisions.

⇒ The study is not a randomised controlled trial and is not powered nor designed to detect causal relationship between EM use and patient outcome.

to be carried out systematically. The function of checklists is to improve task performance; checklists are essentially knowledge-translation tools that promote completeness, consistency and alignment with best practice guidelines when carrying out a task.[1] Checklists work by directing focus, providing facts[2] and guiding—though not dictating—decision-making. Checklists are also believed to improve performance through other mechanisms, such as reducing mental workload, promoting a shared mental model among team members and empowering team members to voice concerns.[3–5]



There are two basic modes of checklist use.[6] In Read-Do mode, checklists are used as the task is performed to guide task completion. During the Do-Confirm mode, checklists are used after task performance to verify completeness. Cognitive aids may even be used to retrieve specific information (for instance, the dosage of a specific medication in a specific setting), so-called Sampling mode.[6 7] Checklists may be used by one or several people, silently or read aloud,[6] and can be categorised according to the circumstances in which they are used: normal (or routine) situations, emergency situations, and atypical (or non-normal) situations.[6 8]

### Checklists for routine situations in healthcare

The WHO Surgical Safety Checklist[9] and the Surgical Safety Patient System[10] are examples of checklists used to improve performance during routine surgical procedures. There is substantial, robust, international evidence that use of the WHO Surgical Safety Checklist reduces mortality and complications associated with general surgery.[11] A checklist designed to minimise bloodstream infections associated with the routine insertion of central venous catheters in the intensive care unit is also associated with improved performance during routine healthcare delivery.[12–14]

### Checklists for healthcare emergencies

Stress affects attention and information retrieval from memory,[15 16] and hence checklists may play a particular role in improving performance during stressful emergencies.[1 4] Due to the unexpected, low-frequency nature of medical emergencies, crisis checklists have mainly been studied using simulation. One study reported a reduction in the frequency of unperformed lifesaving processes of care from 23% to 6% associated with checklist access during simulated surgical crises.[17] Another study reported that checklist use increased the number of key items performed during scenarios in a simulated intensive care unit from 7 to 9.[18] An additional study performed in a simulated surgical ward reported that use of a cognitive aid reduced the frequency of omitted critical management steps from 33% to 10%.[19]

Two studies evaluated crisis checklists in a simulated resuscitation room (RR). One study evaluated the use of an Emergency Protocols Handbook through four scenarios and reported that checklist use decreased error rate from 39% to 19%.[20] The other study evaluated a collection of four checklists through four scenarios; checklist use was associated with an increased frequency of haemodynamic stabilisation and diagnosis recognition, as well as an increased adherence to critical steps (56% without checklist, 82% with checklists).[21]

A simulation-based, randomised controlled trial of medical crisis checklists was performed in four RRs using actual resuscitation teams during their clinical shifts.[22] Eight medical crises were simulated. With checklist access, the median percentage of key interventions performed was 86%; without access, the median percentage was 39%.

Evidence that collections of crisis checklists and cognitive aid elements such as medication doses—emergency manuals (EMs)—improve actual clinical care is limited. Several reports document successful management of perioperative crises thanks to the use of operating room EMs.[23–26] An interview-based study of anaesthesiologists involved in perioperative crises in 69 patients revealed that EM use led to identifying and addressing errors of omission in 59% of cases, while lack of EM use was associated with self-identified errors of omission or delays in intervention performance in 56% of cases.[5] There is also a lack of prospective studies of EM use. One recently published prospective study reported sustained use in the operative setting multiple years after initial implementation.[27]

### Systematic crisis checklist evaluation in the clinical setting

Performance is the end-product of complex interactions between people, the tools at their disposal, and the physical and cultural environments they work in.[28] Checklists do not work in isolation, and their impact on performance is contingent not only on their content and design but also on when, where, how and by whom they are used. Simulation-based trials and interview-based studies[29] provide limited evidence for how checklists may actually impact resuscitation team performance in a specific clinical setting. Several authors have called for pilot studies that systematically evaluate crisis checklists in the clinical setting.[5 19 20 22 29 30]

Experts emphasise the importance of establishing the rationale for EM use, fostering a sense of local EM ownership, ensuring access and familiarity with the tool, and customisation.[31] In particular, an explicit protocol is likely required to ensure that checklists are used as intended. A simulation-based study reported that the mere access to crisis checklists did not ensure their systematic use; only when an assigned team member read aloud the steps on the checklist were all critical actions performed.[32] The Surgical Safety Checklist Implementation Manual emphasises the importance of clearly assigning the role of checklist implementation to a specific person[33] and an interview-based study of EM use during perioperative crises reported beneficial effects of a reader–leader combination.[5]

Challenges to implementation and evaluation of an EM in the emergency department (ED) include the following:

► The number of possible presenting complaints, acute conditions and pre-existing comorbidities among priority 1 patients is large, and the degree of actual acuity is variable.
► The number of personnel that work in the RR is large, and each personnel has variable context-specific degrees of expertise; managing a patient with acute pulmonary oedema may be 'routine' for a seasoned physician and an 'emergency' for a junior resident.
► Do-Confirm mode of checklist use is best suited to quantify the added value of checklist use, while Read-Do and Sampling modes may be best suited for checklist use during an actual crisis.

► Ensuring that checklists relevant to the clinical context are made available to all team members at a suitable juncture during patient management, and encouraging checklist use without negatively impacting on team dynamics, are not self-evident tasks.

These challenges have informed the study design.

## Objectives

The primary objective of the study is to determine whether Do-Confirm EM use leads to the performance, by resuscitation teams managing priority 1 patients in the ED, of additional interventions that are deemed indicated, of neutral relevance or not indicated given the clinical context. Secondary study objectives are

► Determining the frequency and modality (Do-Confirm, Read-Do, Sampling) with which the EM is used and whether these changed significantly over the study period.

► Determining whether Do-Confirm EM use in selected cases where other modes were used might have led to the performance of additional indicated interventions.

► Determining the degree to which resuscitation teams find the EM helpful, and under which circumstances.

## METHODS AND ANALYSIS

### Setting, patient population and personnel

The study will be carried out in the resuscitation room of the ED of Skåne's University Hospital at Lund (referred to hereafter as Lund's RR). Lund's RR is designed for the simultaneous management of three priority 1 patients. EM access during clinical practice will be provided in September 2023. The study population consists of consecutive priority 1 patients managed in the Lund's RR during the 6-month period prior to, and the 6-month pilot study period following EM implementation. Patients are deemed to be priority 1 based on vital signs and symptoms according to the Rapid Emergency Triage and Treatment System (http://predicare.se/) the most commonly used triage system in Sweden. The yearly number of priority 1 patients treated in Lund's RR is roughly 4000 and consists of adult patients (>18 years) with medical or surgical emergencies and patients of all ages with level 1 trauma. Most of these patients are brought to the RR via ambulance, and only occasionally are they identified at the triage or at later stage of ED care. All priority 1 patients are managed immediately in the RR by multidisciplinary resuscitation teams. The core resuscitation team consists of a resident or specialist in emergency medicine, two nurses and one nursing assistant. When the patient is especially critical (for instance, cardiac arrest, level 1 trauma, upper airway obstruction), additional personnel (anaesthesiologists, surgeons, cardiologists, otorhinolaryngologists) supplement the resuscitation team.

### EM: content and format

The EM consists of problem checklists, diagnosis checklists, procedure checklists and fact sheets in digital form.

The checklists and fact sheets are accessed through a tablet computer located at each of the three resuscitation sections in the RR and displayed on a large screen visible to all members of the resuscitation team. Problem checklists consist of lists of interventions and diagnoses to consider based on the patient's presenting symptoms. The popover windows display key clinical features suggestive of specific diagnoses. Diagnosis checklists consist of lists of interventions to consider during the management of a suspected diagnosis. Each intervention is associated with a popover window displaying indications, contraindications or risks associated with the intervention, and details regarding how the intervention is performed, as described in a previous study.[22] Procedure checklists provide step-by-step guidance for the performance of key procedures as well as pictures. Fact sheets provide information such as medical dosages for sedation and anaesthesia for patients of all ages.

The collection of checklists and fact sheets covers the majority of symptoms, diagnoses and situations that may present in the RR, in accordance with the current European Core Curriculum in Emergency Medicine.[34] The format and design of each checklist or fact sheet is informed by the literature pertaining to emergency and abnormal checklists in the aviation industry,[35–37] articles on medical checklists[6 38–40] and a selection of EMs.[41–44] The content is customised to local circumstances in Lund's ED (available medications, medication locations, local routines and phone numbers). EM content and format are further developed through iterative, rapid-cycle simulation-based testing and review by specialists, residents, nurses and nursing assistants who work clinically in Lund's RR. Internet access to the EM is provided to the ED personnel prior to and during the study to allow for familiarisation and suggestions for improvement. Each checklist and fact sheet is approved by at least one specialist in emergency medicine and one nurse, both of whom work clinically in Lund's RR.

### EM: protocol for use

EM use during priority 1 management is strongly advocated but not mandatory. The documenting nurse is assigned the specific task of activating the digital EM. Regarding problem and diagnosis checklists, Do-Confirm is the default mode of EM use—the team first manages priority 1 patients without support from the EM. Once the management plan is finalised (a juncture which may be verbally confirmed with the team leader), the documenting nurse then asks the team: "Which checklist should I display?" When a checklist is displayed, the documenting nurse reads aloud each potential intervention or diagnosis for team consideration. The documenting nurse registers on the digital checklist whether each intervention is

► Deemed indicated and has already been carried out.

► Deemed irrelevant or contraindicated.

► Deemed indicated and will be carried out thanks to EM use.

The physician responsible for patient management has the mandate to request that the EM not be used, to pause or terminate EM use at any time, and to depart from the recommendations featuring on the checklist. Physicians may also request that the EM be displayed directly and used in Read-Do mode or for retrieval of specific information (Sampling mode). Regarding procedure checklists and fact sheets, the default modes of use are Read-Do and Sampling, respectively. After EM use, teams grade the usefulness of the EM on a 1–6 Likert scale prior to logging out. If the documenting nurse logs in to the EM but the EM is not used, the nurse selects among potential reasons for non-use prior to logging out. Teams can also electronically request contact with the investigators to report specific events related to EM use.

### Educational campaign

An educational campaign targeting all personnel who work in Lund's RR precedes the implementation of the EM in clinical practice. Evidence for the potential benefits of an EM—including results from a locally performed crisis checklist study[22]—is presented, as well as the protocol for EM use. Documenting nurses during priority 1 patient management are taught to log in to the EM, display and navigate between checklists, open popover windows, record whether interventions were not indicated, done before or thanks to checklist access, record the team's opinion of the EM's value, request contact with the investigators, and log out. Documenting nurses carry out these functions in addition to their usual documentation tasks.

### Data acquisition

The following data pertaining to each priority 1 patient managed during the 6-month period prior to, and the 6-month pilot study period following implementation of the EM in clinical practice, are extracted from Region Skåne's clinical databases and the national data registry SVAR (Svenska Akutvårdsregistret):

▶ Patient age and sex, presenting complaint and vital signs
▶ Presumptive diagnosis in the RR
▶ Time spent in the RR, in the ED, and admitted to the hospital
▶ Whether the patient was admitted to the intensive care unit, and if so duration of admission
▶ 7-day and 30-day mortality

The following data regarding EM use are recorded on a dedicated computer located within the hospital's firewall each time the EM is activated during the 6-month pilot study period:

▶ Log-in and log-out times
▶ Identification number of the nurse who logged in, level of the most senior physician who was physically present at the bedside during patient management, and patient identification number (which is immediately encrypted)
▶ Checklists and fact sheets that were displayed

▶ Whether interventions were performed before or thanks to checklist access when the EM was used in Do-Confirm mode, or whether the EM was used in Read-Do or Sampling mode instead
▶ The team's subjective evaluation of the value of the EM according to a 1–6 Likert scale
▶ Reasons for EM non-use despite logging in
▶ Whether prompt contact with the research team is requested

The following data are extracted from the medical records of 150 selected cases (see below):

▶ Baseline information regarding the patient's age, sex, known medical background at the time of presentation and current medications
▶ Clinical information regarding the patient's presentation to the RR: ambulance report, history that can be obtained in the RR, vital signs, physical findings, ECG, bedside blood test results and point-of-care ultrasound findings
▶ All interventions (investigations and treatments) performed during the patient's management in the RR
▶ Whether the physician was a specialist or resident, and years of experience
▶ Final hospital diagnosis pertaining to the presentation in the RR

Figure 1 summarises data acquisition during the study period. The primary investigator alone has access to the encryption key. All investigators have access to the anonymised dataset for analysis purposes.

### Impact of EM use on resuscitation team performance

The final 50 cases where the EM was used in Do-Confirm mode and interventions were performed thanks to checklist access are identified for external review. The final cases during the 6-month study period are selected to allow personnel the opportunity to integrate the new tool within their clinical practice. The degree to which each intervention is indicated is independently assessed by three specialists in emergency medicine. All three assessors are blinded to whether the intervention was performed before or after EM use. Two assessors are blinded to the content of the EM, while one is involved in the development and use of the EM. Interventions are categorised, based on the available extracted clinical information, as indicated (eg, administering antibiotics to the septic patient), of neutral relevance (eg, measuring liver function tests for the patient with suspected benzodiazepine overdose), or not indicated (ie, overinvestigation or overtreatment, for example, ordering CT to rule out pulmonary embolism for a patient with pulmonary oedema). When the three assessors are not in agreement regarding the degree of indication of the intervention, resolution is reached through discussion, and based on majority opinion if no consensus can be reached. Management is deemed improved when interventions categorised as indicated are carried out thanks to EM use. EM use is deemed to not add value if additional interventions

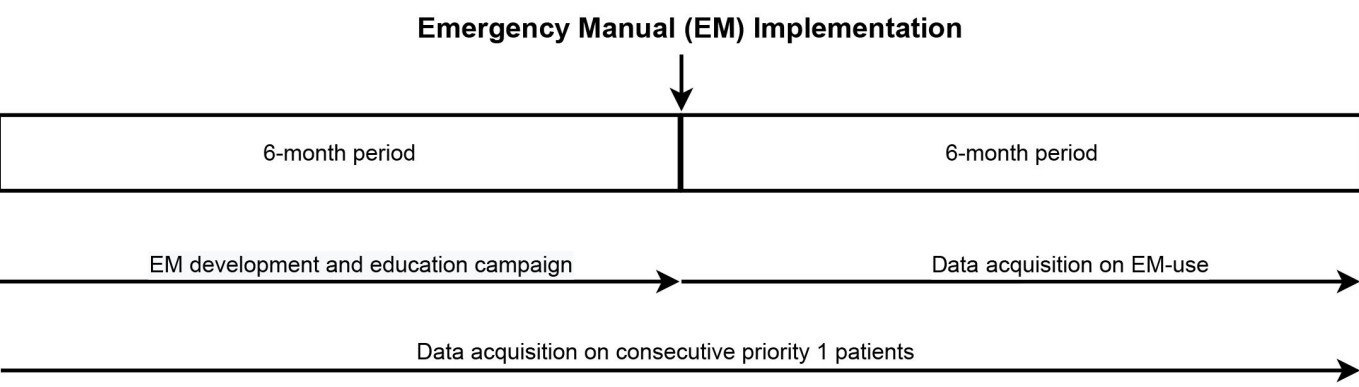

**Figure 1** This figure illustrates the two phases of the study, before and after emergency manual implementation, and how data acquisition corresponds to each phase.

are categorised as of neutral relevance. EM use is deemed detrimental if additional interventions are categorised as not indicated.

### Sample size calculation

There are no published data on EM use during patient management in the ED. We derived a sample of size of 50 patients where at least one intervention was performed thanks to Do-Confirm EM use based on the following assumptions:

► Among patients with at least one intervention performed thanks to the checklist, half of these patients will have one such intervention performed while the other half will have two. Hence, 50 such patients should generate 25+2×25 = 75 interventions thanks to the checklist.

► When multiple checklist-indicated interventions occur in the same patient, the probability of occurrence of these interventions are independent of each other.

In our opinion, a clinically meaningful rate for indicated interventions performed thanks to checklist access is ≥10%. Figure 2 shows that if the true rate of clinically indicated interventions due to the checklist is ≥21% then 50 patients generating 75 interventions will be sufficient to reject the null hypothesis that the indicated rate is ≤10% with at least 80% power and a type I error probability of $\alpha \leq 0.05$.

### Unperformed interventions recommended by the EM

Fifty additional patients where the EM was not used during the pilot study period, and 50 additional patients that presented during the 6-month period preceding EM implementation, are also identified. These 100 patients are selected based on matching to the 50 patients where the EM was used using age, presenting complaint and relevant comorbidities. The EM is accessed to determine whether additional interventions would have

been performed, had the EM been used. All interventions—those actually performed as well as those that might have been performed had the EM been used—are then assessed for degree of indication by three external reviewers as described above. The three specialists are blinded to which interventions were actually performed. Analysing these data may suggest that the EM would have improved management had it been used, with the important caveat that many factors that impact on the suitability of interventions may not be fully conveyed in the patient's chart.

### Specific case descriptions

The quantitative analyses described above are designed to measure EM added value when used in Do-Confirm mode. Yet EMs are arguably designed to be used in Read-Do or Sampling mode in the setting of a true crisis. EM use may also have unforeseen impacts on patient care and team dynamics that warrant prompt remedying. Resuscitations team personnel are encouraged to request prompt contact—via the digital EM—with the research team to report events or concerns relating to EM use during specific situations. The following information is extracted from a structured interview with the personnel regarding the report (online supplemental appendix A):

► Patient age, sex, presenting complaint and suspected diagnosis

► Seniority of the physician initially in charge of the case

► Events or concerns relating to EM use

► Mode of EM use (Do-Confirm, Read-Do, Sampling) and sections relevant to the case

► Personnel's assessment of the impact of the EM on patient care

► Personnel's assessment of impact of the EM on team members and teamwork

► Personnel's suggestions for EM improvement

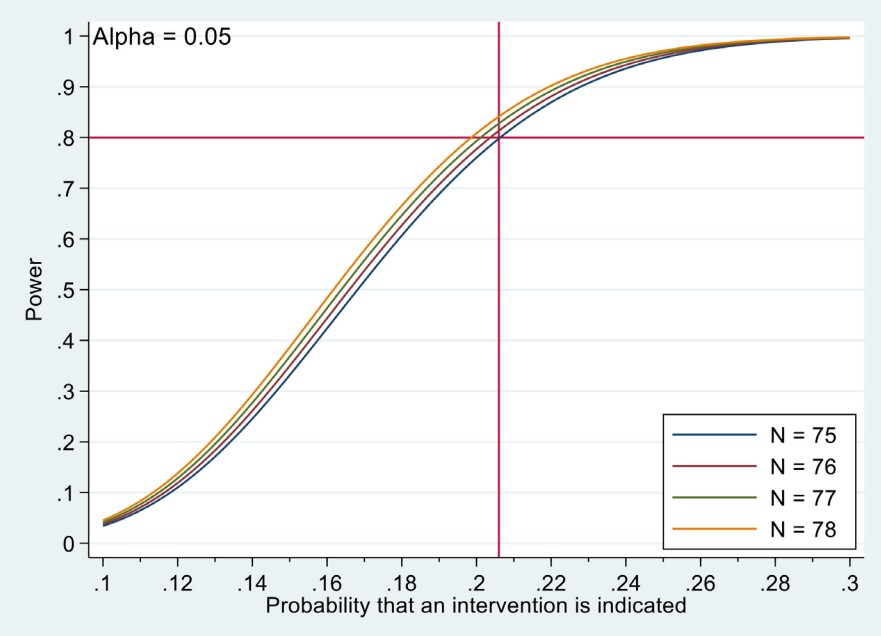

**Figure 2** 'Intervention' refers here to interventions performed by the resuscitation teams thanks to use of the emergency manual. Interventions are graded by external assessors as indicated, neutral or not indicated based on the clinical context. The power curves in the figure plot the power to determine, with p≤0.05, that the proportion of interventions deemed indicated exceeds 10% as a function of the true probability that the intervention is indicated. Assuming that the true proportion of interventions deemed indicated is 21%, 75 interventions are required to meet these requirements. Assuming that half of the patients feature one such intervention and the other half two, then 50 patients can be expected to generate 75 interventions.

## Data analysis

Descriptive statistics are used to analyse EM use data and determine the following (average over and trends throughout the pilot study period):

► Frequency of EM use and mode of use (Do-Confirm, Read-Do or Sampling)
► Which checklists or fact sheets are most frequently accessed
► Time between login to and logout from the digital EM
► Subjective degree of EM value and correlation with specific checklists or fact sheets

Descriptive statistics are also used to determine whether EM use is associated with length of stay in the RR, length of hospital admission and mortality. These analyses are hypothesis-generating and not intended to infer causality. For the 150 patients that are externally reviewed, descriptive statistics are used to report the proportions of interventions deemed indicated, of neutral value or not indicated.

## Patient and public involvement

Patients or the public were not involved in the design of this study, nor will they be involved in its conduct or in the reporting and dissemination of our research.

## ETHICS AND DISSEMINATION
### Ethical and safety considerations

The published literature strongly supports the claim that checklists, used as cognitive aids and not as substitutes for sound clinical judgement, improve performance. Several authors have hypothesised that checklist use may lead to distraction, delays in proper management or wrong management if the wrong checklist is used.[23 45 46] However, published studies have not substantiated these concerns. A simulation-based trial of crisis checklists in Lund's ED did not find any delay in the initiation of patient treatment, and all but one of 15 dangerous or inappropriate interventions were carried out by teams without checklist access.[22] One study reported that EMs caused distraction during a small proportion of perioperative crises without negatively impacting on patient care.[5] According to the study protocol, the physician in charge of patient management has the mandate to terminate EM use at any time, and no constraints are imposed on how the EM is used. For these reasons, patient risk incurred from EM access is deemed minimal.

Priority 1 patients are as a rule severely impaired by an acute critical condition and not in a state where informed consent is possible or appropriate to obtain. Written information about EM use will be available in Lund's RR and provided to patients and relatives on request. Posters informing patients and their relatives about the ongoing trial and how to request non-inclusion will be placed throughout the ED. Patient identification numbers are pseudonymised automatically and data stored on a computer within the hospital's firewall and locked in a room.

No identifying features of the personnel composing the resuscitation teams are registered during the study, aside from the identity of the personnel logging into the

digital EM. Names of personnel involved in the care of specific patients are not recorded during the structured interviews. In the study that evaluated checklists in simulated critical conditions in Lund's RR, over 90% of the care staff stated that they would use the checklist in real crises.[22] Specific reviews of patient management will be based on information available in the medical charts, with the understanding that these cannot convey all relevant factors affecting clinical decisions. No formal complaints or chart reviews will be triggered by the case reviews in this study. For these reasons, risks incurred to resuscitation team personnel from EM access are deemed minimal. The Swedish Ethical Review Authority has approved the study (Dnr 2022-01896-01).

### Dissemination plan
Results will be published in a peer-reviewed journal and abstracts submitted to national and international conferences. The simulation-based study of crisis checklists in four EDs in the Skåne Region reported that >90% of staff at all sites would welcome checklist access during clinical care.[22] Depending on the study results, a multicentre study using customised EMs will be considered.

## STRENGTHS, LIMITATIONS AND SIGNIFICANCE
The literature on medical crisis checklists argues that specific checklists, used according to clear protocols within specific contexts, improve performance. Yet there are no published data on how the introduction of an EM affects the performance of resuscitation teams during the management of actual priority 1 patients in the ED. The main strength of this study is that it evaluates EM use during the management of consecutive, unselected priority 1 patients by resuscitation teams in the ED.

In this study, some of the data on patient characteristics (such as age, chief complaint) and EM use (duration of use, checklists and fact sheets displayed) are automatically registered. Other EM use data (for instance, subjective team evaluation of the EM value, modality of EM use) are actively registered by the documenting nurse. The study is not a randomised controlled trial of EM use, and whether the EM is used or not is at the discretion of each resuscitation team. While these study aspects limit the evaluation of how routine EM use would impact on performance, the study will provide information about the circumstances under which resuscitation teams voluntarily use the EM and deem that it adds value to clinical practice. The study stretches over a 6-month period, allowing for an analysis of how resuscitation teams integrate a new tool within their clinical practice over time.

The study is not powered nor designed to detect causal relationship between EM use and patient outcome. The study will determine whether interventions carried out thanks to checklist use are indicated, of neutral relevance or not indicated based on the clinical context, as assessed by external reviewers. A limitation of this assessment stems from the fact that the external review is based on information available from patient records, which may not convey all the relevant factors affecting clinical decisions.

This study will be conducted in a single academic Swedish ED, limiting the external validity of the results. Yet this study will provide valuable information to guide the development of a tool that has the potential to improve the management of critical patients, and pave the way to multicentre studies of EM use in the ED.

**Contributors** ED conceived of, drafted the original version of, and revised the protocol. POdC, MM, MA and AB reviewed and revised the protocol. WDD performed the sample size calculations, reviewed and revised the protocol. UE conceived of, reviewed and revised the protocol.

**Funding** This work was supported by an ALF research grant (2018-0152) at Skåne University Hospital and by a grant from Region Skåne (2020-0305). This study was part of the AIR Lund (Artificially Intelligent use of Registers at Lund University) research environment and received funding from the Swedish Research Council (VR; grant no. 2019-00198). There was no industry involvement. Funding organisations had no role in the planning, design or realisation of the study, collection, analysis or interpretation of data, or preparation, review or approval of the manuscript.

**Competing interests** None declared.

**Patient and public involvement** Patients and/or the public were not involved in the design, or conduct, or reporting, or dissemination plans of this research.

**Patient consent for publication** Not applicable.

**Provenance and peer review** Not commissioned; externally peer reviewed.

**ORCID iD**
Eric Dryver http://orcid.org/0000-0002-5750-0079

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
