## [Reviewer comments · BMJ Open]

ARTICLE DETAILS

TITLE (PROVISIONAL)	Clinical use of an emergency manual by resuscitation teams and impact on performance in the emergency department—a prospective mixed-methods study protocol
AUTHORS	Dryver, Eric; Olsson de Capretz, Pontus; Mohammad, Mohammed; Armelin, Malin; Dupont, William D.; Bergenfelz, Anders; Ekelund, Ulf

VERSION 1 – REVIEW

REVIEWER	Navalpotro, Susana Emergency Medical Service of Madrid- SUMMA 112, UVI15
REVIEW RETURNED	03-Feb-2023

GENERAL COMMENTS	Congratulations to the authors for the work that is really interesting. Although the purpose of this work is understood, methodologically and as a protocol it would be good to make several changes. Think about the idea is that anyone can replicate your work with the data that are provided. However, in order to provide improvements, some clarifications are indicated below so that this work can be published  [ ] The introduction is clear and very pertinent, but the citations from the bibliography may not have been well selected. Of the 43 citations throughout the manuscript, 26 of them are from the year 2018 or earlier. Please check if the old citations are really relevant, as well as consider some more current ones that may convey the same idea given the large recent bibliography on the subject. [ ] The main objective is not clear. Does the use of the proposed tool lead to better performance? Value a clearer objective. [ ] The cases that seem to have been evaluated in the section Impact of the use of the Emergency Manual on the performance of the resuscitation team, it is not known how these conclusions were reached. [ ] “An additional 50 patients in whom EM was not used during the pilot study period and an additional 50 patients who presented during the six-month period prior to EM implementation are also identified.” The calculation of the sample size is not clear, as is the use of previous cases as a reference and how these are used. If it is a prospective, we understand that the study will be carried out with the cases generated on a particular date and not before. [ ] The voluntary participation of the patients in the study is not clear, nor is it clear how they will obtain their consent. They just report posters in the waiting room with the Project. Perhaps an informed consent protocol could be added for the family or for the patient when they are in a position to do so. [ ] It would be good to add a schedule with project times, as well as the analyzes that will be carried out and the statistics that will be followed to meet the objectives.
------------------	---

	I hope you find these reviews useful, and I encourage you to continue with such interesting and clinically useful work. Cognitive aids are a great tool for an emergency team.
--	--

REVIEWER	Goldhaber-Fiebert, Sara N. Stanford Univ, Anesthesiology
REVIEW RETURNED	08-Feb-2023

GENERAL COMMENTS	Overall and Major Comments: This study by Dryver et al. is novel, highly relevant, and well-written. Most importantly, it contributes to filling a crucial gap in the cognitive aid literature for clinical uses and impacts of emergency manuals (EMs) both for emergency departments and across healthcare settings. Specific positives about the study design, with some small questions/suggestions: +Activating role: Remembering to activate (or trigger) EM use has been a challenge for clinical uses. There's a pro of assigning this to a specific role to help make sure it happens, though, 1) concern for whether the documentation nurse will know which page is most relevant to the patient's event (and if starts on incorrect event page could potentially mislead or confuse team) and 2) whether physician event leader may feel undermined? Suggestion to consider: Multiple hospitals implementing EMs perioperatively, as well as perioperative simulation training programs, have had great success with training nurses (& other interprofessional staff or physicians arriving to help) to ask an 'empowering question' to prompt event leader, such as 'Shall I get the EM?' and if yes (which has almost always been the response) 'To which event?' The documentation nurse could be trained to ask physician event leader some variation of those questions or even skip to the 2nd question if 1st not deemed culturally necessary in your setting. +Assigning reader role to a specific team member has major pros. The main con is if that person is task overloaded. Question: Can documentation nurse reliably do all of: read from EM whenever leader/team need, activate initial use and switch events as needed AND document whenever needed to fulfill both clinical & research documentation roles of 'documentation nurse'? This seems to be a lot, with potentially competing time sensitive priorities. A suggestion to consider is to split out the research documentation (which would not be relevant for future clinical teams) to be a separate research personnel, if funding allows, freeing up the 'documentation nurse' to be 'only' in activating, reader, and clinical documentation roles. +Multiple Design elements including: 1) large format digital display which creates a shared mental model for entire team to see of both which event and specific management steps; 2) auto-recording of whether EM is activated and which sections are displayed; 3) Blinding expert content reviewers to whether EM was used or not. Categories of indicated, neutral, or not indicated. Suggestion: Consider experts assessing also for vital actions which are missing from documented management for that patient's clinical event or situation, i.e. errors of omission? If EM is not yet so broadly accessible locally in pocket/phone formats that clinicians are likely to use it even on non-EM 'control' shifts, then this will be an even cleaner comparison. If mobile versions are already available, ethically their use should not be discouraged for the control group. However, having large, highly visible EM screens accessible and viewable by entire team (vs
------------------	---

	not) would still in my opinion be a more clinically relevant comparison to inform future institutional implementations than defaulting teams to Do-Confirm mode of EM use, which to my knowledge has never been studied in a healthcare context as a sole EM modality and has substantial limitations as described by authors and above. Major Study Design change to consider: As detailed below, suggest that If ethically and practically feasible, authors consider randomizing by shift (or 24 hour day) for the large format digital EM to be accessible or not, with further details and rationale described below. This would require a new power calculation and parallel changes to abstract and throughout methods, though in my opinion would be a much stronger study. Design would be somewhat similar to prior Dryver et. al. simulation-based study (BMJ Qual Safety, 2021 and Arriaga NEJM 2013, regarding training for all plus randomization to accessible EM or not) and in this case the impacts of EM implementation and flexible use could be much better assessed. P15 L47 “Yet EMs are arguably designed to be used in Read-Do or Sampling mode in the setting of a true crisis.” This thoughtful statement by authors raises an important concern which I share: The impact on patient management and on team communication with current study design may be artificially influenced by the research team requesting that EM should by default be used specifically in Do-Confirm mode, even with an option to deviate or ‘opt out’ from this. On P8 L27 authors state “Do-Confirm mode of checklist use is best suited to quantify the added value of checklist use, while Read-Do and Sampling modes may be best suited for checklist use during an actual crisis” Defaulting teams to ‘Do-Confirm’ mode (versus allowing whatever modality serves leader/team best for that patient’s crisis management) does allow for data collection of which actions were explicitly prompted by EM use, as authors point out. However, my major concern is that this very limited mode of use may not measure actual EM impacts across flexible modalities, and would be particular to this research study, potentially creating an artificial context. Moreover, if many teams choose to use EM more flexibly than Do-Confirm, that would be a meaningful sign of uptake, but would then further impair the current study design, given there is no randomized control group in current design. If no positive impact were found, is that because of the substantially limited modality? And if a positive impact were found, is that generalizable to broader modalities that would not naturally be defaulted to Do-Confirm in non-research clinical settings? i.e. Will this research data be generalizable? In my opinion: Even if some team leaders may already have access to pocket or mobile versions of the EM from prior work at this institution, which would ethically not be reasonable to ban on control shifts or days (though potentially could be noted by research personnel), the large digital highly visible versions and activation/reader role and institutional encouragement of team use are likely to nudge the intervention group to have substantially more as well as more impactful EM use. If the proposed study design changes are practically or ethically not feasible, I do think that there are sufficient strengths and that this still study does help to fill some important gaps as currently
--	--

	designed in an arena of clinical EM uses that has been hard to study. Further Suggested Design Details (if changing study design):  -Trainings for all to use EM, when accessible, in whatever modality serves that team and crisis best. These could be similar to the trainings you describe, though instead of defaulting to Do-Confirm suggested use, would teach all three types of helpful EM uses (Read-Do; Do-Confirm; Sampling) with examples of when a specific mode of use may be more useful than others. -EM accessible, vs EM not accessible, randomized per shift. -Teams may use or not use EM, in any modality, based on their clinical judgment and assessment of available personnel. Measure also, as already proposed: EM uses in each of the modalities and non-uses, and if able collect info after for non-uses on why EM was not used. -If possible, separate out the research documentation role to be a person separate from the documentation & EM activation and EM reader nurse role. The latter person would reasonably need to juggle EM roles along with other clinical documentation roles for a fair comparison to EM non-use, in order for future clinical teams to not require an extra person. However, adding the research documentation roles to same person may create task overload, negatively impacting either clinical documentation and EM reader role or research documentation. The data from this revised study design would be more applicable to non-research clinical situations than encouraging only Do-Confirm EM use because it would include EM benefits as well as any distractions or harms, from all three use modalities (Read-Do, Do-Confirm, and Sampling). Additionally, for both EM accessible and not accessible shifts, actions taken can be assessed by experts as authors already describe. The specialists, blinded to EM accessibility or not, could then do their described review of interventions. This substantial study design change would clearly have impacts for all methods sections. Other Detailed Suggestions: P2 L10 Suggest adding the word 'clinical' in front of 'perioperative crises,' to more accurately describe that study. "An interview-based study suggests that use of an emergency manual (EM)—a collection of crisis checklists—improves management of CLINICAL perioperative crises." P2 L29 (& P12 top): Is the documentation nurse an extra (research) personnel? If this person also has clinical documentation/timing/other clinical tasks in addition to EM activator, reader, and documenter, this may likely create task overload, with some tasks necessarily dropped (either research or clinical). Please clarify and talk with some nurses in these clinical roles to assess whether it is actually feasible for one person to do all these tasks well the majority of the time. P5 L15 "Checklists work by directing focus and providing facts [2]" Suggest adding something like: and guide, though don't dictate decision-making. Suggest adding something like: Emergency manuals contain elements of checklists while also including broader cognitive aid elements e.g. sampling information for key medication dosing.
--	---

	P5 L26 “In Read-Do mode, checklists are used as the task is performed.” Suggest adding “...to guide task completion.” This will also make it have parallel language and structure to the next sentence. P5 L 31 “Cognitive aids may even be used in a non-systematic manner to retrieve specific information” – suggest changing non-systematic, which sounds like a negative judgment, to “as needed” or a similar neutral term. P5 L59 Suggest minor changes in this sentence to prevent 2 uses of the word checklist in same sentence and to emphasize the patient outcomes that Refs 12-14 detail (vs ‘performance’) “A checklist designed to minimize bloodstream infections associated with the insertion of central venous catheters in the intensive care unit is also associated with improved patient outcomes during routine health care delivery.” P6 L29 near or after REFS 17 and 18 that discuss sim studies evidence for emergency checklists, suggest adding/describing also recent study by: P7 L12: Nuanced word difference: suggest changing ‘identifying’ to ‘catching’ or something similar: “...EM use led to identifying errors...” In that study, the delays/errors of omission were not only identified but rather real-time caught (or identified and addressed) by the clinicians based on using EM, which may get lost by using solely the word identifying. P9 Terminology clarification here and throughout: Suggest changing ER to RR. When reading closely, I understand and appreciate the definitions starting Line 15 of ED vs ER. However 1) the E in ER does not make sense as an abbreviation for resuscitation. 2) Readers may be confused by apparent mixed use of ED and ER in the paragraphs that follow. 3) In the United States at least, the term for the entire area used to be called ER, which was usually part of the Department of Surgery at an academic medical center, and as this discipline became more respected and became its own department, it was renamed at most places as ED. Abbreviating resuscitation room as RR (instead of ER) would avoid this confusion. P11 L19: “The documenting nurse then displays the checklist(s) relevant to the specific circumstances and reads aloud each potential intervention/diagnosis for team consideration.” More clarity is needed here regarding who decides which checklist(s) is relevant to the specific circumstances? When to switch between checklists (e.g. when new information is available or evolving patient status, or anything else that changes the most helpful ‘page’ to be on). Often that is a physician team-leader rather than an EM reader task. See major comments above regarding interactions of these roles and empowering question. P11 L40: “Physicians may also request that the EM be displayed directly and used in Read-Do mode or for retrieval of specific information (Sampling mode).”
--	--

	It's good for safety that the responsible physician may use these other modes (or terminate EM use or depart from recommendations). This section would benefit from a mention that measurements will include when and which mode(s) are used (as authors detail later, on P13 L3). See also major study design comment above. P13 L36: Clarification of clinical responsibilities at study institution: "whether the physician was a specialist or resident..." Does this refer to the primary physician leader for this crisis or the most senior 'responsible' physician for the shift? Are there cases in which a resident is completely in charge of resuscitation room management without any more senior backup help available? P14 L15 in detail: Please define terms, particularly 'of neutral relevance' and 'not indicated'? In the context of a crisis, all actions take valuable time and attention which are resources in short supply for timely completion of other 'as indicated' actions. E.g. Are actions 'of neutral relevance' if there is controversy in the literature and 'not indicated' if unrelated to managing that situation? Or are actions 'of neutral relevance' if unrelated to managing that situation and 'not indicated' if potentially harmful in that situation (meaning beyond the time and attention they take away from other tasks)? If the latter definitions are the intent, authors may also consider 'not applicable' and 'harmful' as potentially clearer terms for second and third categories. Either way, and most importantly terms should be defined. -P15 L15 describes comparing cases with EM use to matched cases during the study period in which EM is not used. Does that mean that teams may choose not to use it, for various reasons (versus not accessible on some shifts)? Based upon later details, I presume the former, but if keeping current study design this should be made more clear on P15 (P19, L22: "The study is not a randomized controlled trial of EM use, and whether the EM is used or not is at the discretion of each resuscitation team."). -Incomplete Reference 16 to fill in details - Consider integrating this relatively recent study into Intro or Discussion, as one of the few not mentioned RCTs outside of perioperative setting. Koers L, van Haperen M, Meijer CG, van Wandelen SB, Waller E, Dongelmans D, Boermeester MA, Hermanides J, Preckel B. Effect of cognitive aids on adherence to best practice in the treatment of deteriorating surgical patients: a randomized clinical trial in a simulation setting. JAMA surgery. 2020 Jan 1;155(1):e194704-.
--	--

VERSION 1 – AUTHOR RESPONSE

REVIEWER 1

Mrs. Susana Navalpotro, Emergency Medical Service of Madrid- SUMMA 112, Autonomous University of Madrid

Congratulations to the authors for the work that is really interesting. Although the purpose of this work is understood, methodologically and as a protocol it would be good to make several changes. Think about the idea is that anyone can replicate your work with the data that are provided. However, in order to provide improvements, some clarifications are indicated below so that this work can be published

1-Citations

The introduction is clear and very pertinent, but the citations from the bibliography may not have been well selected. Of the 43 citations throughout the manuscript, 26 of them are from the year 2018 or earlier. Please check if the old citations are really relevant, as well as consider some more current ones that may convey the same idea given the large recent bibliography on the subject.

1-Authors' response

This study focusses on evaluating crisis checklists in the Emergency Department. Studies and publications of highest relevance are those focusing on crisis checklists in health care fields. Some studies focusing on the use of checklists in general are relevant for the sake of providing context.

Regarding studies published as of 2019: we have added the study by Koers et al 2020 at the suggestion of the second reviewer (ref 19). We have also added a reference to the publication by Kapadia et al (2020) calling for cognitive aid evaluation in the clinical setting (ref 29). We have carried out a repeat literature review and not found additional studies pertaining specifically to crisis checklists within the ED. If we have missed relevant studies, we would be grateful for suggestions!

Regarding studies published prior to 2019: we believe that our references are either directly related to the focus of the study (e.g. ref 1, Goldhaber-Fiebert et al. *Anesth Analg* 2013;117(5):1149-61; or ref 17, Arriaga et al. *N Engl J Med* 2013;368(3):246-53) or provide valuable context about the use of checklists (e.g. ref 9, Haynes et al. *N Engl J Med* 2009;360(5):491-9). We welcome suggestions for deleting or replacing specific references.

2-Main Objective

The main objective is not clear. Does the use of the proposed tool lead to better performance? Value a clearer objective.

2-Authors' response

Thank you for this suggestion. The text in the manuscript (P8 L20) is as follows: "The primary objective of the study is to determine whether Do-Confirm EM use leads to the performance, by resuscitation teams managing priority 1 patients in the ED, of additional interventions that are deemed indicated, of neutral relevance or not indicated given the clinical context." The term 'Do-Confirm' is described on P5 L13. To further clarify, we have added examples of what we mean by "indicated", "of neutral relevance" and "not indicated" on P14 L11-14.

If our study reveals that EM use leads to the performance of additional interventions that are "indicated", this would indicate that EM use leads to better performance.

3-Case Selection

The cases that seem to have been evaluated in the section Impact of the use of the Emergency Manual on the performance of the resuscitation team, it is not known how these conclusions were reached.

3-Authors' response

The text in the manuscript under the section "Impact of Emergency Manual use on resuscitation team performance" is as follows: "The final 50 cases where the EM was used in Do-Confirm mode and

interventions were performed thanks to checklist access are identified for external review." We are selecting cases used in Do-Confirm mode because those are the cases for which we can identify interventions that would not have been carried out had the EM not been available.

4-"Control Cases" and Sample Size Calculation

"An additional 50 patients in whom EM was not used during the pilot study period and an additional 50 patients who presented during the six-month period prior to EM implementation are also identified." The calculation of the sample size is not clear, as is the use of previous cases as a reference and how these are used. If it is a prospective, we understand that the study will be carried out with the cases generated on a particular date and not before.

4-Authors' response

The text in the manuscript regarding sample size calculation is as follows (P14 L23):

"We derived a sample of size of 50 patients where at least one intervention was performed thanks to Do-Confirm EM use based on the following assumptions:

- Among patients with at least one intervention performed thanks to the checklist, half of these patients will have one such intervention performed while the other half will have two. Hence, 50 such patients should generate $25 + 2 \times 25 = 75$ interventions thanks to the checklist.
- When multiple checklist-indicated interventions occur in the same patient, the probability of occurrence of these interventions are independent of each other.

In our opinion, a clinically meaningful rate for indicated interventions performed thanks to checklist access is $\geq 10\%$. Figure 2 shows that if the true rate of clinically indicated interventions due to the checklist is $\approx 21\%$ then 50 patients generating 75 interventions will be sufficient to reject null hypotheses that the indicated rate is $\leq 10\%$ with at least 80% power and a Type I error probability of $\alpha \leq 0.05$."

We are unsure of how to clarify this text further, but welcome specific suggestions.

We will also use two matched patient groups consisting of 50 patients to look for errors of omission that might have been addressed had the EM been used:

- 1-The first group consists of patients treated after EM implementation without EM use
- 2-The second group consists of patients treated during the 6-month period prior to EM implementation.

How these groups are used is described in the manuscript (P15 L17):

"The EM is accessed to determine whether additional interventions would have been performed, had the EM been used. All interventions—those actually performed as well as those that might have been performed had the EM been used—are then assessed for degree of indication by three external reviewers as described above. The three specialists are blinded to which interventions were actually performed. Analysing this data may suggest that the EM would have improved management had it been used, with the important caveat that many factors that impact on the suitability of interventions may not be fully conveyed in the patient's chart."

Again, suggestions as to how this text can be made clearer are very welcome.

5-Informed Consent

The voluntary participation of the patients in the study is not clear, nor is it clear how they will obtain their consent. They just report posters in the waiting room with the Project. Perhaps an informed consent protocol could be added for the family or for the patient when they are in a position to do so.

5-Authors' response

As the reviewer alludes to, it is not ethically justifiable nor often feasible to obtain informed consent from patients treated in the resuscitation room. The following text features in the manuscript (P18 L1):

"Priority 1 patients are as a rule severely impaired by an acute critical condition and not in a state where informed consent is possible or appropriate to obtain."

This study focuses on knowledge translation. We are not introducing a new treatment but providing a cognitive aid to help teams deliver established treatments. The literature on medical checklists has so far failed to show that checklists negatively impact on health care. Our previous study (Dryver et al. Medical crisis checklists in the emergency department: a simulation-based multi-institutional randomised controlled trial. *BMJ Qual Saf* 2021;30(9):697-705), showed that checklists do not impede the delivery of first line measures and that checklist access reduces the delivery of inappropriate interventions. Based on the above, we argued in the application to The Swedish Ethical Review Authority that obtaining informed consent from each patient was neither feasible nor necessary from an ethical standpoint. Our study meets the definition of a pragmatic clinical trial. Sugarman and Califf *JAMA*. 2014;311(23):2381-2 argue that in such studies ... "obtaining conventional written informed consent may be not only ethically unnecessary but may render such research impracticable because of logistical burdens and may introduce selection bias." The Swedish Ethical Review Authority has approved the study protocol (Dnr 2022-01896-01).

6-Schedule

It would be good to add a schedule with project times, as well as the analyzes that will be carried out and the statistics that will be followed to meet the objectives.

6-Authors' response

Regarding the schedule, the following text features in the manuscript (P9 L10):

"EM access during clinical practice will be provided in September 2023. The study population consists of consecutive priority 1 patients managed in the Lund RR during the six-month period prior to, and the six-month pilot study period following EM implementation."

Regarding the analyses and the statistics, the following text features in the manuscript (P16 L19):

"Descriptive statistics are used to analyse EM use data and determine the following (average over and trends throughout the pilot study period):

- frequency of EM use and mode of use (Do-Confirm, Read-Do, or Sampling)
- which checklists or fact sheets are most frequently accessed
- time between login to and logout from the digital EM
- subjective degree of EM value and correlation with specific checklists or fact sheets

Descriptive statistics are also used to determine whether EM use is associated with length of stay in the RR, length of hospital admission and mortality. These analyses are hypothesis-generating and not intended to infer causality. For the 150 patients that are externally reviewed, descriptive statistics are used to report the proportions of interventions deemed indicated, of neutral value or not indicated."

We are unsure of how to clarify this text further, but welcome specific suggestions.

Conclusion Remarks

I hope you find these reviews useful, and I encourage you to continue with such interesting and clinically useful work. Cognitive aids are a great tool for an emergency team.

Authors' response

Thank you for your comments and support for this study! We agree that the study has direct bearing on clinical work and that cognitive aids have great potential. As we have written above, we welcome suggestions as to how the references can be made more relevant and the text improved for clarity.

REVIEWER 2

Overall and Major Comments: This study by Dryver et al. is novel, highly relevant, and well-written. Most importantly, it contributes to filling a crucial gap in the cognitive aid literature for clinical uses and impacts of emergency manuals (EMs) both for emergency departments and across healthcare settings. Specific positives about the study design, with some small questions/suggestions:

1-Activating role

Remembering to activate (or trigger) EM use has been a challenge for clinical uses. There's a pro of assigning this to a specific role to help make sure it happens, though, 1) concern for whether the documentation nurse will know which page is most relevant to the patient's event (and if starts on incorrect event page could potentially mislead or confuse team) and 2) whether physician event leader may feel undermined? Suggestion to consider: Multiple hospitals implementing EMs perioperatively, as well as perioperative simulation training programs, have had great success with training nurses (& other interprofessional staff or physicians arriving to help) to ask an 'empowering question' to prompt event leader, such as 'Shall I get the EM?' and if yes (which has almost always been the response) 'To which event?' The documentation nurse could be trained to ask physician event leader some variation of those questions or even skip to the 2nd question if 1st not deemed culturally necessary in your setting.

1-Authors' response

We thank the reviewer for this suggestion and we have changed the text (Emergency Manual: protocol for use) in the manuscript to the following on P11 L9: "The documenting nurse then asks the team: "Which checklist should I display?" When a checklist is displayed, the documenting nurse reads aloud each potential intervention/diagnosis for team consideration." Adding this empowering question will hopefully decrease the risk that an irrelevant checklist is displayed, and decrease the risk that EM use leads to conflicts between the documenting nurse and physician event leader.

Some additional comments:

1-We agree that selecting the pertinent checklist(s) is essential and not necessarily straightforward. Based on our experience with simulations, we believe that displaying the checklist on a large screen for all team members minimizes the risk that an irrelevant checklist is used. Our checklists contain information that are helpful for all team-members, including specific instructions as to where medications can be found and how they are administered. We therefore believe that addressing the empowering question to the whole team is appropriate.

2-We recognize that checklist use may create conflict between the documenting nurse and the physician event leader. The empowering question will hopefully be as successful in our institution as in those the reviewer refers to. We hope that conflicts can further be avoided by explicitly empowering physicians to curtail EM use, depart from the checklist, or request that the EM be used up-front in Read-Do or Sampling mode.

3-We provide teams with the ability to promptly contact the study investigators to discuss specific issues related to EM use. Using the structured interview form, the study investigator will specifically enquire about:

- a) Assessed impact of EM use on patient care (clinical decisions regarding diagnosis, treatment)
- b) Assessed impact of EM use on team members/teamwork (organization, communication)

We hope that potential interpersonal conflicts regarding EM-use can then be promptly addressed.

2-Separate research personnel

Assigning reader role to a specific team member has major pros. The main con is if that person is task overloaded. Question: Can documentation nurse reliably do all of: read from EM whenever leader/team need, activate initial use and switch events as needed AND document whenever needed to fulfill both clinical & research documentation roles of 'documentation nurse'? This seems to be a lot,

with potentially competing time sensitive priorities. A suggestion to consider is to split out the research documentation (which would not be relevant for future clinical teams) to be a separate research personnel, if funding allows, freeing up the 'documentation nurse' to be 'only' in activating, reader, and clinical documentation roles.

2-Authors' response

We believe that the documenting nurse can manage his/her clinical duties as well as the EM for the following reasons:

1-The majority of priority 1 patients in our institution are managed by teams consist of "a resident or specialist in emergency medicine, two nurses and one nursing assistant." There are therefore three health care personnel that can carry out measures while the documenting nurse documents interventions and manages the EM.

2-The recommended default mode of EM-use is Do-Confirm. According to this mode of use, urgent measures requiring the help of the documenting nurse would be performed prior to EM use.

3-We have striven to minimize the amount of research documentation required from the documenting nurse. Which pages of the EM that are displayed are automatically registered. The documenting nurse needs to log in to the EM, navigate to the relevant checklist(s), click on boxes linked to specific measures, select from 1 to 6 for EM-use evaluation, select whether contact with investigators is requested, and log-out. All these steps entail clicking on boxes, circles or numbers aside from the log-in step where 6 numbers need to be entered.

4-We did not observe that EM-management impeded team clinical performance during our in-situ simulation study (Dryver et al 2021 BMJ Qual Saf Figure 3). Granted, this study did not include logging in and out of the EM, navigating between checklists and clicking on boxes.

To include consecutive patients, the study will be on-going 24/7 for 6 months and will include priority 1 patients that are managed simultaneously in the resuscitation room by separate teams. Having separate research personnel is not feasible, nor do we believe that it is necessary.

3-Errors of omission

Multiple Design elements including: 1) large format digital display which creates a shared mental model for entire team to see of both which event and specific management steps; 2) auto-recording of whether EM is activated and which sections are displayed; 3) Blinding expert content reviewers to whether EM was used or not. Categories of indicated, neutral, or not indicated. Suggestion: Consider experts assessing also for vital actions which are missing from documented management for that patient's clinical event or situation, i.e. errors of omission?

3-Authors' response

We were inspired by the reviewer's article in *Anesth Analg* 2020;131(6):1815-1826. We will use two groups of 50 matched patients to look for errors of omission (P15 L14): "Fifty additional patients where the EM was not used during the pilot study period, and 50 additional patients that presented during the six-month period preceding EM implementation, are also identified. These 100 patients are selected based on matching to the 50 patients where the EM was used using age, presenting complaint and relevant co-morbidities. The EM is accessed to determine whether additional interventions would have been performed, had the EM been used. All interventions—those actually performed as well as those that might have been performed had the EM been used—are then assessed for degree of indication by three external reviewers as described above. The three specialists are blinded to which interventions were actually performed. Analysing this data may suggest that the EM would have improved management had it been used, with the important caveat

that many factors that impact on the suitability of interventions may not be fully conveyed in the patient's chart."

Clearly, the expert physicians might also identify—for all cases—measures deemed indicated that do not feature in the EM. This will provide an opportunity to further develop the EM.

4-EM Access/Display

If EM is not yet so broadly accessible locally in pocket/phone formats that clinicians are likely to use it even on non-EM 'control' shifts, then this will be an even cleaner comparison. If mobile versions are already available, ethically their use should not be discouraged for the control group. However, having large, highly visible EM screens accessible and viewable by entire team (vs not) would still in my opinion be a more clinically relevant comparison to inform future institutional implementations than defaulting teams to Do-Confirm mode of EM use, which to my knowledge has never been studied in a healthcare context as a sole EM modality and has substantial limitations as described by authors and above.

4-Authors' response

Physicians in our institution use a variety of cognitive aids, some of which are in printed format and others accessible on-line. Our previous study showed that the use of an EM specifically tailored to local circumstances and displayed for the whole team leads to significantly more indicated interventions performed within 15 min, despite all personnel having access to their usual cognitive aids.

The issues of randomizing teams to EM access or not and the Do-Confirm as default EM mode of use are addressed below.

5-Randomization

Major Study Design change to consider: As detailed below, suggest that If ethically and practically feasible, authors consider randomizing by shift (or 24 hour day) for the large format digital EM to be accessible or not, with further details and rationale described below. This would require a new power calculation and parallel changes to abstract and throughout methods, though in my opinion would be a much stronger study. Design would be somewhat similar to prior Dryver et. al. simulation-based study (BMJ Qual Safety, 2021 and Arriaga NEJM 2013, regarding training for all plus randomization to accessible EM or not) and in this case the impacts of EM implementation and flexible use could be much better assessed.

5-Authors' response

We considered such a study design but opted for the current proposal for the following reasons:

1-We believe that familiarizing personnel with the EM and its use, and then depriving personnel of EM access every other day or shift, may be detrimental to patient care. Personnel may assume that certain checklists or facts (e.g. the dosing of intranasal fentanyl for a child with burns) will be available and not mentally prepare for situations where other cognitive aids are required. It is ethically problematic to develop and deliver a tool designed to help manage crises (e.g. a fire-extinguisher as analogy), teach the use of the tool, promote its use, and then deprive teams of the tool every other day.

2-The EM is a new tool and one of the objectives of the proposed study is to evaluate how teams integrate a tool that is available point-of-care 24/7 into their clinical practice. We anticipate that adoption and integration of the EM into practice will take time. Having the tool inaccessible every other shift would significantly impede this study objective.

3-To evaluate the value-added of EM-use, we will identify patients where the EM was used in Do-Confirm mode. In this way, the patient acts as his/her own control. Instead of analysing patients treated every other day without EM access, we will analyse 50 matched patients treated during the 6-month prior to EM implementation. We will also analyse 50 matched patients treated post EM-implementation without the use of the EM. We thereby will use three approaches to analyse the value added by EM-use.

6-Do-Confirm

P15 L47 "Yet EMs are arguably designed to be used in Read-Do or Sampling mode in the setting of a true crisis." This thoughtful statement by authors raises an important concern which I share: The impact on patient management and on team communication with current study design may be artificially influenced by the research team requesting that EM should by default be used specifically in Do-Confirm mode, even with an option to deviate or 'opt out' from this. On P8 L27 authors state "Do-Confirm mode of checklist use is best suited to quantify the added value of checklist use, while Read-Do and Sampling modes may be best suited for checklist use during an actual crisis" Defaulting teams to 'Do-Confirm' mode (versus allowing whatever modality serves leader/team best for that patient's crisis management) does allow for data collection of which actions were explicitly prompted by EM use, as authors point out. However, my major concern is that this very limited mode of use may not measure actual EM impacts across flexible modalities, and would be particular to this research study, potentially creating an artificial context. Moreover, if many teams choose to use EM more flexibly than Do-Confirm, that would be a meaningful sign of uptake, but would then further impair the current study design, given there is no randomized control group in current design. If no positive impact were found, is that because of the substantially limited modality? And if a positive impact were found, is that generalizable to broader modalities that would not naturally be defaulted to Do-Confirm in non-research clinical settings? i.e. Will this research data be generalizable?

6-Authors' response

In accordance with the reviewer's concerns, we have changed the text on P11 L7 to: "We recommend Do-Confirm as the default mode of EM use." In defence of the current protocol:

1-We would not characterize Do-Confirm as a "very limited mode of use." Most physicians do not require a checklist to order crystalloid infusion, cultures and antibiotics for a patient with septic shock and we anticipate that, for the majority of priority one patients, teams will start treatment prior to using the EM as they have been doing for many years. The EM is a new tool, and until teams are familiar with using the tool in the clinical context, it is wise for teams to manage priority one patients as usual, carrying out urgent, first-line therapies first, and then accessing the EM once the patient has been stabilized to determine whether additional interventions are indicated.

2-As per current modification, we will recommend Do-Confirm as the default mode of use, but "Physicians may also request that the EM be displayed directly and used in Read-Do mode or for retrieval of specific information (Sampling mode)." One aim of the study is to determine how teams use the EM as their familiarity with the tool grows. As mentioned in the Data analysis section, we plan to specifically evaluate trends in frequency of EM use and mode of use during the 6-month study period. While Do-Confirm mode of use allows for identification and evaluation of additional interventions, observing that teams use the EM according to Read-Do and Sampling (and rate the EM as valuable using the Likert score) at an increasing rate would provide evidence that EM-use is deemed valuable.

3-There is no randomized controlled study group, but we will use 100 matched patients where the EM was not used as controls to look for errors of omission (see 5-Authors' response). In addition, 50 patients where ≥ 1 additional measure was performed thanks to EM-use in Do-Confirm mode are

used to evaluate the value of additional measures. Using Do-Confirm, the patient is his/her own control.

4-This will be a single-centre pilot study, which per se limits the generalizability of the results. Yet we believe that data regarding how the EM is used during the end of the 6-month period will provide valuable information about EM use in a clinical as opposed to an "artificial" context. By merely recommending Do-Confirm as default mode of use and explicitly allowing for Read-Do and Sampling, we will not prevent teams from using the EM as they see fit within the clinical context.

7-Value of Display

In my opinion: Even if some team leaders may already have access to pocket or mobile versions of the EM from prior work at this institution, which would ethically not be reasonable to ban on control shifts or days (though potentially could be noted by research personnel), the large digital highly visible versions and activation/reader role and institutional encouragement of team use are likely to nudge the intervention group to have substantially more as well as more impactful EM use.

7-Authors' response

We agree. We believe that the use of the large screens to display checklists and fact sheets for the whole team improves performance through several mechanisms:

1-It promotes a shared mental model

2-It promotes "cross-checking" whereby all team-members are aware that other team-members also see the checklist

3-The previously vetted checklists and fact sheets convey the notion that performing indicated measures is endorsed by the institution

8-Value of current study design

If the proposed study design changes are practically or ethically not feasible, I do think that there are sufficient strengths and that this still study does help to fill some important gaps as currently designed in an arena of clinical EM uses that has been hard to study.

8-Authors' response

We are relieved to read the above paragraph! We believe that intermittent checklist access would negatively impact on uptake of a new tool and integration into daily practice, that depriving teams of a crisis-management tool that is simultaneously promoted is contradictory, potentially hazardous and ethically problematic.

9-Training

Further Suggested Design Details (if changing study design): Trainings for all to use EM, when accessible, in whatever modality serves that team and crisis best. These could be similar to the trainings you describe, though instead of defaulting to Do-Confirm suggested use, would teach all three types of helpful EM uses (Read-Do; Do-Confirm; Sampling) with examples of when a specific mode of use may be more useful than others.

9-Authors' response

Modes of use (Do-Confirm, Read-Do and Sampling) are brought up during the training sessions in our institution during which personnel are familiarized with the EM's structure and content. The EM contains fact sheets (e.g. options for pediatric PSA) where Sampling is the only mode of use. At the suggestion of the reviewer, we will explicitly provide examples of when certain modes of use may be more advantageous than others. It is worth pointing out in this context that physicians/teams have varying degrees of competence/experience regarding specific clinical situations and that the optimal mode of use is not solely determined by the patient's presenting problem or diagnosis.

10-EM accessible, vs EM not accessible, randomized per shift.

-Teams may use or not use EM, in any modality, based on their clinical judgment and assessment of available personnel. Measure also, as already proposed: EM uses in each of the modalities and non-uses, and if able collect info after for non-uses on why EM was not used.

10-Authors' response

We thank the reviewer for this valuable suggestion. Should the documenting nurse log in to the EM and the EM not be used, he/she will be prompted to select among a short menu of potential reasons for non-EM use (instead of clicking on team evaluation of the EM) prior to logging out. We have added the following sentence under the Protocol section, P11 L20: "If the documenting nurse logs in to the EM but the EM is not used, the nurse selects among potential reasons for non-use prior to logging out." We have added the following under the Data acquisition section, P13 L9: "reasons for EM non-use despite logging in". We do not have a realistic strategy to document reasons for total EM non-use (no log-in) that would work 24/7 over 6-months given that our primary tool for data capture is using the EM.

According to the current protocol, "frequency of EM use and mode of use (Do-Confirm, Read-Do, or Sampling)" are measured (P16 L21)

11-Separate research person

If possible, separate out the research documentation role to be a person separate from the documentation & EM activation and EM reader nurse role. The latter person would reasonably need to juggle EM roles along with other clinical documentation roles for a fair comparison to EM non-use, in order for future clinical teams to not require an extra person. However, adding the research documentation roles to same person may create task overload, negatively impacting either clinical documentation and EM reader role or research documentation.

11-Authors' response

We do not anticipate that managing the EM will lead to task overload (please see 2-Authors' response). Recruiting 1 or more research personnel 24/7 over a 6-month period is not feasible. One of the goals of the study is to determine how current resuscitation teams integrate a new tool into their clinical practice. Having a research person present might impact of team behaviour through the Hawthorn effect and create an "artificial" setting.

12-Revised Study Design

The data from this revised study design would be more applicable to non-research clinical situations than encouraging only Do-Confirm EM use because it would include EM benefits as well as any distractions or harms, from all three use modalities (Read-Do, Do-Confirm, and Sampling). Additionally, for both EM accessible and not accessible shifts, actions taken can be assessed by experts as authors already describe. The specialists, blinded to EM accessibility or not, could then do their described review of interventions. This substantial study design change would clearly have impacts for all methods sections.

12-Authors' response

We have changed the wording (P11 L7) to "recommend Do-Confirm as default mode of use" and will convey more nuanced recommendations for EM use to the teams. Please see 6-Authors' response. The specialists blinded to EM use will assess 100 cases where the EM was not used but where interventions suggested by the EM are included for assessment of degree of indications. Please see 3-Authors' response.

13-Adding 'clinical'

P2 L10 Suggest adding the word 'clinical' in front of 'perioperative crises,' to more accurately describe that study. "An interview-based study suggests that use of an emergency manual (EM)—a collection of crisis checklists—improves management of CLINICAL perioperative crises."

13-Authors' response

We have changed the sentence in the abstract to: "An interview-based study suggests that use of an emergency manual (EM)—a collection of crisis checklists—improves management of clinical perioperative crises."

14-Task overload

P2 L29 (& P12 top): Is the documentation nurse an extra (research) personnel? If this person also has clinical documentation/timing/other clinical tasks in addition to EM activator, reader, and documenter, this may likely create task overload, with some tasks necessarily dropped (either research or clinical). Please clarify and talk with some nurses in these clinical roles to assess whether it is actually feasible for one person to do all these tasks well the majority of the time.

14-Authors' response

We did not observe any task overload during the study we carried out in-situ in our RR. The nurses responded on the questionnaire that they would use the tool in a clinical situation if it were available. Granted, this study did not include logging in and out of the EM, navigating to the desired checklist and clicking on boxes, but we do not believe that these additional tasks will result in task overload. Please see 2-Authors' response.

We have clarified that the documenting nurse is not an extra (research) personnel on P12 L9: "Documenting nurses carry out these functions in addition to their usual documentation tasks."

We have also replaced "documentation nurse" with "documenting nurse" x 4 to avoid confusion arising from different terminology.

15-Mechanism of action of checklists & EM-content

P5 L15 "Checklists work by directing focus and providing facts [2]' Suggest adding something like: and guide, though don't dictate decision-making. Suggest adding something like: Emergency manuals contain elements of checklists while also including broader cognitive aid elements e.g. sampling information for key medication dosing.

15-Authors' response

We have modified the sentence on P5 L7 to: "Checklists work by directing focus, providing facts [2], and guiding—though do not dictating—decision-making."

We have modified the sentence on P7 L4 to: "Evidence that collections of crisis checklists and cognitive aid elements such as medication doses—Emergency Manuals (EM)—improve actual clinical care is limited."

16-Adding "to guide task completion"

P5 L26 "In Read-Do mode, checklists are used as the task is performed." Suggest adding "...to guide task completion." This will also make it have parallel language and structure to the next sentence.

16-Authors' response

We have modified the sentence on P5 L12 to: "In Read-Do mode, checklists are used as the task is performed to guide task completion."

17-"Non-systematic"

P5 L 31 "Cognitive aids may even be used in a non-systematic manner to retrieve specific information" – suggest changing non-systematic, which sounds like a negative judgment, to "as needed" or a similar neutral term.

17-Authors' response

We have removed "in a non-systematic manner." The sentence on P5 L14 now reads: "Cognitive aids may even be used to retrieve specific information (for instance the dosage of a specific medication in a specific setting), so-called Sampling mode."

18-Preventing use of the "checklist" x 2

P5 L59 Suggest minor changes in this sentence to prevent 2 uses of the word checklist in same sentence and to emphasize the patient outcomes that Refs 12-14 detail (vs 'performance') "A checklist designed to minimize bloodstream infections associated with the insertion of central venous catheters in the intensive care unit is also associated with improved patient outcomes during routine health care delivery."

18-Authors' response

We have made the recommended change.

19-Additional Simulation Study

P6 L29 near or after REFS 17 and 18 that discuss sim studies evidence for emergency checklists, suggest adding/describing also recent study by:

19-Authors' response

We thank the reviewer for bringing this study to our attention. We have added the reference to the study by Koers et al in the following sentence on P6 L13: "An additional study performed in a simulated surgical ward reported that use of a cognitive aid reduced the frequency of omitted critical management steps from 33% to 10% [19]." The context implies that the simulations dealt with medical crises.

20-Modifying "Identified"

P7 L12: Nuanced word difference: suggest changing 'identifying' to 'catching' or something similar: "...EM use led to identifying errors..." In that study, the delays/errors of omission were not only identified but rather real-time caught (or identified and addressed) by the clinicians based on using EM, which may get lost by using solely the word identifying.

20-Authors' response

We have changed the sentence on P7 L7 to the following: "An interview-based study of anaesthesiologists involved in perioperative crises in 69 patients revealed that EM use led to identifying and addressing errors of omission in 59% of cases, while lack of EM use was associated with self-identified errors of omission or delays in intervention performance in 56% of cases [5]."

21-RR instead of ER

P9 Terminology clarification here and throughout: Suggest changing ER to RR. When reading closely, I understand and appreciate the definitions starting Line 15 of ED vs ER. However 1) the E in ER does not make sense as an abbreviation for resuscitation. 2) Readers may be confused by apparent mixed use of ED and ER in the paragraphs that follow. 3) In the United States at least, the term for the entire area used to be called ER, which was usually part of the Department of Surgery at an academic medical center, and as this discipline became more respected and became its own department, it was renamed at most places as ED. Abbreviating resuscitation room as RR (instead of ER) would avoid this confusion.

21-Authors' response

We have replaced ER with RR throughout the manuscript.

22-Which Checklist is Relevant

P11 L19: "The documenting nurse then displays the checklist(s) relevant to the specific circumstances and reads aloud each potential intervention/diagnosis for team consideration." More clarity is needed here regarding who decides which checklist(s) is relevant to the specific circumstances? When to switch between checklists (e.g. when new information is available or evolving patient status, or anything else that changes the most helpful 'page' to be on). Often that is a physician team-leader rather than an EM reader task. See major comments above regarding interactions of these roles and empowering question.

22-Authors' response

We believe that adding the empowering question at the reviewer's suggestion addresses this issue. The question is addressed to the team in general, in accordance with the CRM-principle of implementing team-based tools. Whether and when to switch between checklists will depend on the specifics of the case. Some checklists have as starting point the patient's main problem (e.g. low SpO2), others have as starting point a suspected diagnosis (e.g. pulmonary edema), some fact sheets provide pictures of procedures. Several checklists may be relevant to the case. We do not believe that generic statements regarding switching between checklists add value to the manuscript.

23-Adding to the section that mode of EM use will be measured

P11 L40: "Physicians may also request that the EM be displayed directly and used in Read-Do mode or for retrieval of specific information (Sampling mode)." It's good for safety that the responsible physician may use these other modes (or terminate EM use or depart from recommendations). This section would benefit from a mention that measurements will include when and which mode(s) are used (as authors detail later, on P13 L3). See also major study design comment above.

23-Authors' response

The section on P11 describes the protocol for EM use. Starting on P12, there is a section entitled "Data acquisition" where it states that the following is recorded (P13 L5): "whether interventions were performed before or thanks to checklist access when the EM was used in Do-Confirm mode, or whether the EM was used in Read-Do or Sampling mode instead." Due to constraints in the size of the manuscript, we have chosen to not repeat this information in the section that deals with the protocol for EM use.

24-Seniority of physician

P13 L36: Clarification of clinical responsibilities at study institution: "whether the physician was a specialist or resident..." Does this refer to the primary physician leader for this crisis or the most senior 'responsible' physician for the shift? Are there cases in which a resident is completely in charge of resuscitation room management without any more senior backup help available?

24-Authors' response

In the Data acquisition section on P13 L1, it states: "level of the most senior physician who was physically present at the bedside during patient management." This would be the lead physician during the shift if he/she is physically present at the bedside during patient management but not otherwise. There are cases where residents manage resuscitations without the physical presence of senior colleagues, but backup help is always one phone call away.

25-Definitions of "Neutral relevance" and "Not indicated"

P14 L15 in detail: Please define terms, particularly 'of neutral relevance' and 'not indicated? In the context of a crisis, all actions take valuable time and attention which are resources in short supply for

timely completion of other 'as indicated' actions. E.g. Are actions 'of neutral relevance' if there is controversy in the literature and 'not indicated' if unrelated to managing that situation? Or are actions 'of neutral relevance' if unrelated to managing that situation and 'not indicated' if potentially harmful in that situation (meaning beyond the time and attention they take away from other tasks)? If the latter definitions are the intent, authors may also consider 'not applicable' and 'harmful' as potentially clearer terms for second and third categories. Either way, and most importantly terms should be defined.

25-Authors' response

One concern with checklist-use is that it may lead to over-investigation and over-treatment:

- checklists based on the patient's presenting problem (e.g. low SpO₂) provide lists of potential diagnoses to consider, with key suggestive bedside information (e.g. massive PE, consider if RV dilation on ultrasound). The use of this checklist may lead to over-investigations (e.g. CT-PA in a clinical context highly suggestive of pulmonary edema).
- checklists based on the patient's diagnosis (e.g. angioedema) provide lists of potential treatments to consider, along with indications and contraindications/risks. The use of these checklists may lead to over-treatment (e.g. push-dose pressor use in a septic patient with a systolic blood pressure of 90 mm Hg).

For 150 patients, we will ask three expert physicians to grade all the investigations and treatments the patient received—prior to EM use, thanks to EM use, and when the EM was not used, those that would have delivered had the EM been used) for the degree of indication, based on the clinical context provided by the patient chart. To clarify the terminology, we modified the text on P14 L10 to the following: "Interventions are categorized, based on the available extracted clinical information, as indicated (e.g. administering antibiotics to the septic patient), of neutral relevance (e.g. measuring liver function tests for the patient with suspected benzodiazepine overdose), or not indicated (i.e. over-investigation or -treatment, e.g. ordering computed tomography to rule out pulmonary embolism for a patient with pulmonary edema).

We cannot provide more detailed definitions for "indicated", "of neutral relevance" and "not-indicated". The degree of indication will be determined through expert consensus based on the available clinical context, knowledge of the medical literature and clinical experience. There is a subjective component to the assessment, which will apply similarly to all interventions—those carried out with and without EM-use.

Regarding the suggestion to the use the term "harmful": the checklists are vetted by several personnel in our institution and do not feature items that are on face-value harmful. Rather, items of the checklist may be carried out when the clinical context suggests that these items are not indicated. As the reviewer mentioned above, checklists guide—but do not dictate—management.

26-Control cases where EM not used

-P15 L15 describes comparing cases with EM use to matched cases during the study period in which EM is not used. Does that mean that teams may choose not to use it, for various reasons (versus not accessible on some shifts)? Based upon later details, I presume the former, but if keeping current study design this should be made more clear on P15 (P19, L22: "The study is not a randomized controlled trial of EM use, and whether the EM is used or not is at the discretion of each resuscitation team.").

26-Authors' response

The reviewer's interpretation of the text is correct. EM-use cannot be made mandatory given the current level of evidence for its benefit. We have added the following sentence at the beginning of the section Emergency Manual: protocol for use on P11 L6: "EM use during priority 1 management is

strongly advocated but not mandatory." The catch-22 is that evidence for the benefit of the EM cannot be obtained unless the tool is used, and repeated use is required to establish under which circumstances the EM adds value.

27-Incomplete reference 16

Incomplete Reference 16 to fill in details

27-Authors' response

We have filled in missing information regarding the publication by Dismukes et al. 2015 (ref 16).

28-Additional reference

Consider integrating this relatively recent study into Intro or Discussion, as one of the few not mentioned RCTs outside of perioperative setting. Koers L, van Haperen M, Meijer CG, van Wandelen SB, Waller E, Dongelmans D, Boormeester MA, Hermanides J, Preckel B. Effect of cognitive aids on adherence to best practice in the treatment of deteriorating surgical patients: a randomized clinical trial in a simulation setting. JAMA surgery. 2020 Jan 1;155(1):e194704-.

28-Authors' response

We thank the reviewer for bringing this study to our attention. We have added the reference to the study by Koers et al in the following sentence on P6 L13: "An additional study performed in a simulated surgical ward reported that use of a cognitive aid reduced the frequency of omitted critical management steps from 33% to 10% [19]."

Concluding remarks

Feel free to contact me with any questions.

Respectfully,

Sara Goldhaber-Fiebert, MD (she/her)

saragf@stanford.edu

Clinical Professor of Anesthesiology, Perioperative & Pain Medicine

Stanford University School of Medicine

Authors' response

Thank you for all that you have contributed to improving this manuscript, and for your support for this study!

VERSION 2 – REVIEW

REVIEWER	Goldhaber-Fiebert, Sara N. Stanford Univ, Anesthesiology
REVIEW RETURNED	01-Apr-2023

GENERAL COMMENTS	Thank you for this excellent revision and your impactful work that is exciting to see happening. The authors should be commended for addressing all major comments thoughtfully, with an appropriate combination of changes as well as relevant explanations for why not to make some changes (eg overall study design). Clearly, they have been considering the many challenges in this research area, alternative approaches, and their pros and cons. This study will help to fill a major gap in the literature for implementation and clinical use of cognitive aids for rare crises. In particular, the following elements will make novel contributions to thorny challenges across multiple disciplines seeking to implement and
------------------	--

use EMs effectively: activating use systematically via a specific role and empowering question; assigned reader role; large screen display for team viewing and shared mental model during crises.

Below are some substantive comments that I hope will make your work even more impactful, as well as a few minor details. While I'm sure they will consider and address each comment thoughtfully, if the authors disagree conceptually with any comment, with their reasoning, I still recommend acceptance.

Logistics:

-Numbers (alone) refer to my original comment number (and authors' response with same #).

-Letters refer to new comments.

-Page & Line numbers (as P_, L_) refer to the clean version of this revision, given that Page 1 in authors' PDF starts there. The numbers are the larger numbers closer to text, as I see two sets of numbers on each page.

-For convenience and clarity, specific wording suggestions are provided where changes are recommended, though authors should feel free to edit wording as they see fit, to make sure the nuances of their study are appropriately communicated.

a. P2, L6 Abstract: "To date, there are no prospective studies of EMs during clinical practice."

Our team (very recently) published a prospective, large observational study of perioperative EM use and sustainment, that is complementary to your study design. That study specifically assessed peri-crisis uses, with denominators of ~25,000 cases per study periods and also denominators of crises with cardiac arrest to measure use rates. After an initial (expected) drop at one year post-implementation, it found sustainment of meaningful rates of use six years later.

Note: I would have mentioned this in my prior review, though it was still embargoed after acceptance then, and it is now published online at Journal of Clinical Anesthesia. Your current study takes a deeper dive into specific cases, with a systematic team-based approach, which is much needed.

P7, L10: Suggest describing our relevant just published study, e.g. by adding a sentence at the end of the section on 'Checklists for healthcare emergencies' Addition of something like: "Studies of emergency manual clinical use rates, that include case denominators, are just beginning, though there is early evidence for sustainment of clinical uses multiple years after initial implementation." REF BELOW (versus elsewhere in the introduction, as appropriate).

See: <https://doi.org/10.1016/j.jclinane.2023.111111>

(or author note: if that does not give you access yet for any reason; You can also use for the next few months this free shareable-link that takes you directly to PDF:

<https://authors.elsevier.com/a/1gquY3OfKxo%7Eea>

Citation: Goldhaber-Fiebert SN, Frackman A, Agarwala AV, Doney A, Pian-Smith MCM.

Emergency Manual Peri-Crisis Use Six Years Following Implementation: Sustainment of an Intervention for Rare Crises. J Clin Anes. Aug 2023. doi.org/10.1016/j.jclinane.2023.111111 (Publisher Note: Print page numbers will be added later, though citation is available online already as above)

The challenges of activating use, reader role, and shared screen with shared mental model are all thorny practical issues for which your study will add meaningful novel value across all disciplines of EM use, so that is more the aspect I would recommend emphasizing, though there are certainly very few prospective studies. While I would consider your study here to be the first to my knowledge of a prospective in-depth study of specific EM uses and non uses, and would happily describe it myself as such a year from now, I always hesitate to use the word 'first' or 'no prior' at the time of publication, given the possibilities of overlap in timing with other studies that none of us yet know about.

In summary for this point: Recommend removing the P2, L6 quoted above, or changing it to something like "There is a need for in-depth prospective studies of EM uses during clinical practice and in particular addressing challenges of activating EM use, enabling effective team use, and supporting a shared mental model."

6 Do-Confirm: The authors responded: 'In accordance with the reviewer's concerns, we have changed the text on P11 L7 to: "We recommend Do-Confirm as the default mode of EM use."'

I'm concerned that my prior comments had an unintended consequence and that this change has created a new issue. My comments about use mode had all been intended in the broader realm of why to consider a completely different study design, with randomization by shift for 'cleaner' results. However, the authors have carefully considered and well-explained their study design decision for why NOT to randomize by shift. I'm convinced.

So, zooming out for a bigger picture view: within the (reasonably selected) study design of NOT randomizing, there are likely more cons than pros to emphasizing to entire team a single default mode of use (e.g. Do-Confirm as default recommendation). There are already multiple potential barriers to activating EM use, including:

- remembering to activate use later (prospective memory being one of the most vulnerable forms of memory)
- defining when a nurse should activate use after "The team first manages priority 1 patients without support from the EM..." and will use be early enough? See more below
- healthcare cultures that shame providers for cognitive aid use (especially in past though still occurs), with a risk here to unintentionally add a new (study design) barrier to early use. Especially more junior or new providers may be hesitant to suggest appropriate EM use that is against the 'default' protocol.
- variety of modes of helpful EM uses seen in past simulation studies and clinical uses, as the authors describe, (often depending on the event, team experience with that event, type of checklist etc), as authors well understand.

I would strongly favor the following: NOT recommending to the physician lead nor team of providers which default type of use is desired and as in clinical practice, leaving that decision to the leader/team. However, the documenting nurse would be trained on when to activate use (in Do-Confirm mode, which may be either the

	initial use or may systematically build upon and fill in from prior selective EM uses). This study has so much positive to contribute and will be essentially training local providers in how to use EMs. My biggest concern is that stating a strong default to Do-Confirm mode, for study design purposes, could potentially make providers hesitant to ask for an EM before 'proving' their full cognitive knowledge, which would cause backsliding in culture and training. Whether or not authors agree with this suggestion or any variation of it (which I leave to them ultimately, given they have the most knowledge of design and their setting), please make sure to check the entire paper for consistency. Eg Page 10, Line 12 "Procedural checklists provide step-by-step guidance" does not make sense in the current manuscript with Do-Confirm default, if any procedure were done before EM activation. The thoughtful ways that authors have built in the empowering question from EM nurse and the large screen for shared viewing will help activate appropriate use in Do-Confirm mode if there was no EM use yet (or interrupted use). Per study protocol, the documenting nurse will be trained to ask that empowering question at a certain point in patient stability. To clarify, I would suggest still keeping this Do-Confirm mode for standardized EM activation and training of documenting nurse, just not training entire team to a default Do-Confirm mode for the reasons above. Note: How to identify when to activate EM use should be further defined in both methods here and for nurse training protocol, as it is often unclear or forgotten when 'then' is. For data analysis, if most leaders/teams do not prompt use separately prior to that, this will achieve the Do-Confirm intended 'default' study design, with patients serving as their own controls as the authors state. And if many leaders/teams choose to initiate use earlier than prompted by documenting nurse, eg actively requesting EM for sampling mode eg of a medication detail or for Read-Do mode in the case of a very unfamiliar event, those cases would show smaller, if any, quantitative changes for the impact of EM on clinically appropriate actions, after EM is used to systematically check actions later. However, as the authors initially suggested and I agree: the #/% of those cases would make their own important statement about the value clinicians see in the tool, and the other Do-Confirm cases could still be analyzed in the initial ways described. Question: Will you have or can you think of a way to capture whether the nurse activated EM in Do-Confirm mode (later) by asking an empowering question about which checklist to display? Versus whether a leader/team member requested a specific checklist (earlier; sampling or Read-Do eg procedural checklist or unfamiliar event for those clinicians)? If so, could consider taking the 50 cases from the former group for a cleaner study. If not, it would seem reasonable still to compare actions before and after EM use, whether that use is activated by documenting nurse or whether that use is requested early by a team member for sampling of key information or Read-Do mode for a procedure etc. b. P11, L7-10 Emergency Manual Protocol for Use
--	--

	“We recommend Do-Confirm as the default mode of EM use—the team first manages priority 1 patients without support from the EM. The documenting nurse then asks the team: “Which checklist should I display?” -The bigger comment above suggests that tweaking language here to something like “After team begins patient management with initial actions, the documenting nurse then asks the team: “Which checklist should I display?” -In addition, more detail would be relevant here to define when is ‘then’ and how will the documenting nurse be triggered to recognize and therefore activate EM use with this empowering question. During an evolving medical crisis, as opposed to a pre-procedure setup, there is no clear moment when ‘Do’ management is completed and ‘Confirm’ checking begins. Both briefly for the methods, and then for in more detail for the nurse training, this will be important to define more clearly. It may be helpful to read if not familiar with/cite here if space allows (and/or in subsequent study article) ‘implementation intentions’ literature for If/Then behavioral triggers, developed originally by Gollwitzer PhD and applied to healthcare: Classic/Broad References: Gollwitzer P, Sheeran P. Implementation intentions and goal achievement: A metaanalysis of effects and processes. Adv Exp Soc Psychol. 2006;38:69–119. or Gollwitzer PM. Implementation intentions: Strong effects of simple plans. Am Psychol. 1999;54:493–503. &/or Reference an overview of their application to healthcare, somewhat more recently, eg: Saddawi-Konefka D, Schumacher DJ, Baker KH, Charnin JE, Gollwitzer PM. Changing physician behavior with implementation intentions: closing the gap between intentions and actions. Academic Medicine. 2016 Sep 1;91(9):1211-6. Some ideas to consider for prompting ‘If/Then’ in the nurses’ training: -If chest compressions have begun but no event has been announced -If noise level is increasing (chaotic environment) -If actions are being repeated but patient is not improving -If get to a certain time point from patient arrival to trauma bay How you choose If/Then triggers for nurse activation of EM use will be influenced by both ideal EM use, human factors for remembering to trigger use, and this research study design. For the combination of these, given the nurse is also documenting time presumably, it strikes me that the ‘certain time point from patient arrival to trauma bay’ may be a good way to both trigger nurse activation and to standardize. Authors would know best for deciding in ED RR, what amount of time that would be for the sweetspot goal of not too early to interrupt potentially effective team management and not too late to help with life-saving omissions or course corrections. Minor Points:
--	--

	P5, L7 Typo: “and guiding—though do not dictating—decision-making.” Remove the ‘do’ P7, L22 Typo: ‘rational’ should be ‘rationale’ (noun) P8, L7-16 “Challenges” bullets. Recommend adding (for implementation challenges):  -Activating use -Enabling shared team use, including a reader, with shared mental model, e.g. via large display If you wish, it would be relevant to expand slightly on each here, as mentioned above in the novel contribution description.
--	---

VERSION 2 – AUTHOR RESPONSE

REVIEWER:

Thank you for this excellent revision and your impactful work that is exciting to see happening. The authors should be commended for addressing all major comments thoughtfully, with an appropriate combination of changes as well as relevant explanations for why not to make some changes (eg overall study design). Clearly, they have been considering the many challenges in this research area, alternative approaches, and their pros and cons. This study will help to fill a major gap in the literature for implementation and clinical use of cognitive aids for rare crises. In particular, the following elements will make novel contributions to thorny challenges across multiple disciplines seeking to implement and use EMs effectively: activating use systematically via a specific role and empowering question; assigned reader role; large screen display for team viewing and shared mental model during crises.

Below are some substantive comments that I hope will make your work even more impactful, as well as a few minor details. While I’m sure they will consider and address each comment thoughtfully, if the authors disagree conceptually with any comment, with their reasoning, I still recommend acceptance.

Logistics:

- Numbers (alone) refer to my original comment number (and authors’ response with same #).
- Letters refer to new comments.
- Page & Line numbers (as P_, L_) refer to the clean version of this revision, given that Page 1 in authors’ PDF starts there. The numbers are the larger numbers closer to text, as I see two sets of numbers on each page.
- For convenience and clarity, specific wording suggestions are provided where changes are recommended, though authors should feel free to edit wording as they see fit, to make sure the nuances of their study are appropriately communicated.

a. P2, L6 Abstract: “To date, there are no prospective studies of EMs during clinical practice.” Our team (very recently) published a prospective, large observational study of perioperative EM use and sustainment, that is complementary to your study design. That study specifically assessed peri-crisis uses, with denominators of ~25,000 cases per study periods and also denominators of crises with cardiac arrest to measure use rates. After an initial (expected) drop at one year post-implementation, it found sustainment of meaningful rates of use six years later.

Note: I would have mentioned this in my prior review, though it was still embargoed after acceptance then, and it is now published online at Journal of Clinical Anesthesia. Your current study takes a deeper dive into specific cases, with a systematic team-based approach, which is much needed.

AUTHORS RESPOND:

Thank you for bringing your recently published study of EM use—based on 75,000 operative cases!—to our awareness. Your paper is now referenced in our revision.

REVIEWER:

P7, L10: Suggest describing our relevant just published study, e.g. by adding a sentence at the end of the section on 'Checklists for healthcare emergencies' Addition of something like: "Studies of emergency manual clinical use rates, that include case denominators, are just beginning, though there is early evidence for sustainment of clinical uses multiple years after initial implementation." REF BELOW (versus elsewhere in the introduction, as appropriate).

AUTHORS RESPOND:

We have added the following two sentences at the end of the section on 'Checklists for healthcare emergencies': "There is also a lack of prospective studies of EM use. One recently published prospective study reported sustained use in the operative setting multiple years after initial implementation {Goldhaber-Fiebert, 2023 #3803}." P7 L10

REVIEWER:

The challenges of activating use, reader role, and shared screen with shared mental model are all thorny practical issues for which your study will add meaningful novel value across all disciplines of EM use, so that is more the aspect I would recommend emphasizing, though there are certainly very few prospective studies. While I would consider your study here to be the first to my knowledge of a prospective in-depth study of specific EM uses and non uses, and would happily describe it myself as such a year from now, I always hesitate to use the word 'first' or 'no prior' at the time of publication, given the possibilities of overlap in timing with other studies that none of us yet know about.

AUTHORS RESPOND:

We agree with your point.

1-We have removed the sentence in the abstract reading "To date, there are no prospective studies of EMs during clinical practice"

2-The strengths and limitations of this study section featured the following: "To date, studies of Emergency Manuals (EM)—collections of crisis checklists— have been simulation- or interview-based." We have replaced the sentence with: "To date, the majority of studies of Emergency Manuals (EM)—collections of crisis checklists— have been simulation-based, with the notable exceptions of one interview-based[1] and one prospective study[2]. (P4 L2)

REVIEWER:

In summary for this point: Recommend removing the P2, L6 quoted above, or changing it to something like "There is a need for in-depth prospective studies of EM uses during clinical practice and in particular addressing challenges of activating EM use, enabling effective team use, and supporting a shared mental model."

AUTHORS RESPOND:

We have replaced the deleted sentence in the abstract with the following: "There is a need for in-depth prospective studies of EM use during clinical practice, evaluating when and how EMs are used and impact on patient management." P2 L6

We hesitate to write that the specific goals of the study are to address the challenges of activating EM use, enabling effective team use, and supporting a shared mental model. Previous literature on

checklists has highlighted the challenge of initiating checklist use and the benefit of an assigned reader. Thanks to your suggestion during the previous review, we have now protocolized an empowering question to prompt EM use. Yet the study does not specifically focus on these challenges, e.g. by randomizing teams to reader/no-reader, empowering question vs no empowering question.

It is not clear how the EM that will be evaluated during the study is most effectively used. Effective use may depend not only on the patient's signs/symptoms/suspected diagnosis but also on the experience and competence of the physician leading the team and the degree to which the team-members are familiar with the EM itself. We anticipate that the study will generate valuable information that will guide EM development and training regarding its use.

The use of a large screen to display the checklists for the whole team is consistent with CRM's promotion of team-based tool development and implementation. We surmise that displaying checklists on a large screen will promote a shared-mental model among team members. Yet the study is not designed to assess to what degree a shared mental model is achieved through the use of the large screens.

REVIEWER:

6 Do-Confirm

The authors responded: 'In accordance with the reviewer's concerns, we have changed the text on P11 L7 to: "We recommend Do-Confirm as the default mode of EM use."' I'm concerned that my prior comments had an unintended consequence and that this change has created a new issue. My comments about use mode had all been intended in the broader realm of why to consider a completely different study design, with randomization by shift for 'cleaner' results. However, the authors have carefully considered and well-explained their study design decision for why NOT to randomize by shift. I'm convinced.

So, zooming out for a bigger picture view: within the (reasonably selected) study design of NOT randomizing, there are likely more cons than pros to emphasizing to entire team a single default mode of use (e.g. Do-Confirm as default recommendation). There are already multiple potential barriers to activating EM use, including:

- remembering to activate use later (prospective memory being one of the most vulnerable forms of memory)
- defining when a nurse should activate use after "The team first manages priority 1 patients without support from the EM..." and will use be early enough? See more below
- healthcare cultures that shame providers for cognitive aid use (especially in past though still occurs), with a risk here to unintentionally add a new (study design) barrier to early use. Especially more junior or new providers may be hesitant to suggest appropriate EM use that is against the 'default' protocol.
- variety of modes of helpful EM uses seen in past simulation studies and clinical uses, as the authors describe, (often depending on the event, team experience with that event, type of checklist etc), as authors well understand.

I would strongly favor the following: NOT recommending to the physician lead nor team of providers which default type of use is desired and as in clinical practice, leaving that decision to the leader/team. However, the documenting nurse would be trained on when to activate use (in Do-Confirm mode, which may be either the initial use or may systematically build upon and fill in from prior selective EM uses).

AUTHORS RESPOND:

We agree with the reviewer that there is a difference between stating that a given mode of use is the

default mode versus stating that a certain mode of use is recommended.

The wording in the original manuscript was: "The default mode of EM use is Do-Confirm." The wording in the revision 1 was: "We recommend Do-Confirm as the default mode of EM use-." We have changed the sentence to the following: "Regarding problem and diagnosis checklists, Do-Confirm is the default mode of EM use." (P11 L13)

There are two main arguments for Do-Confirm as default mode of EM use in the proposed study.

1-We are introducing a new tool into a complex environment. SEIPS (Systems Engineering Initiative for Patient Safety; e.g. Holden RJ, Carayon P. SEIPS 101 and seven simple SEIPS tools. *BMJ Qual Saf* 2021) provides a framework to highlight the complex interactions between personnel, tools, tasks and environment. We do not yet know how the EM will impact on team performance. Given the large number of personnel working in the resuscitation room, it is logistically impossible to provide in-situ simulation-based training to familiarize all personnel with EM-use in the clinical context. During the initial phase of the study period, personnel will be less familiar with the content of the EM and how it can bolster team-performance. Currently, patients are managed without crisis checklists displayed for the whole team. A cautious implementation of the EM consists in teams managing patients as usual initially, and accessing/using the EM once the team-leader no longer has suggestions regarding further management (i.e. Do-Confirm as default mode of use).

2-Our previous study showed that EM use improved the management of simulated patients with selected conditions. However, we do not have the data required to advocate that the EM be used up-front as Read-Do during clinical practice. The only way to acquire data that supports the hypothesis that checklist use leads to the performance of additional indicated measures is for the checklist to be used in Do-Confirm mode.

According to the study protocol, the physician has the mandate to request use of the EM at any point during patient management, and we anticipate that as the physicians involved in the study become more familiar with the EM, the use of the EM may shift from mainly Do-Confirm to Read-Do. Changing the wording from "recommended mode of use" to "default mode of use" will avoid sending mixed signals.

The main function of the default mode of use is to prevent conflict between the physician team-leader and the documenting nurse. In 2008 WHO Implementation Manual Surgical Safety Checklist (First edition), it reads: "A possible disadvantage of having a single person lead the Checklist is that an antagonistic relationship might be established with other operating team members. The Checklist coordinator can and should prevent the team from progressing to the next phase of the operation until each step is satisfactorily addressed, but in doing so may alienate or irritate other team members."

The absence of default mode of EM use could lead to the documenting nurse interrupting or impeding appropriate patient management by suggesting that the EM be used in Read-Do mode, leading to a conflict between the documenting nurse and the physician leading the team. We reason that it is in the interest of patient safety to introduce the EM with Do-Confirm as default mode, i.e. that the documentation nurse asks an empowering question once the physician has run out of ideas regarding patient management (see additional comments below regarding timing).

REVIEWER:

This study has so much positive to contribute and will be essentially training local providers in how to use EMs. My biggest concern is that stating a strong default to Do-Confirm mode, for study design purposes, could potentially make providers hesitant to ask for an EM before 'proving' their full cognitive knowledge, which would cause backsliding in culture and training.

AUTHORS RESPOND:

Physicians belong to different specialties, and the pool of physicians who manage patients in the resuscitation room is large. While the EM will be available on-line to allow physicians (especially residents and specialists in emergency medicine working in the ED) to familiarize themselves with the EM's content, it is not logistically possible to introduce all physicians to how the EM may be used in clinical practice through in-situ simulation-based training. The optimal mode of EM use may depend on the physician's knowledge and degree of experience. Do-Confirm mode of EM use may be most efficient for some physicians/teams while Read-Do mode may be better for others. The study may show that certain checklists end-up being used mostly in one mode or another. Instead of stating that "This study . . . will be essentially training local providers in how to use EMs", we would state that the study will familiarize local providers to an EM within the context of clinical care and record how its use evolves during a six-month study period.

We hope that the use of the term "default" will not hinder physicians from using the EM in Sampling or Read-Do. There is at present no culture of, nor training in, EM use in our department. The study will no doubt generate information that will guide further EM development and training regarding use.

REVIEWER:

Whether or not authors agree with this suggestion or any variation of it (which I leave to them ultimately, given they have the most knowledge of design and their setting), please make sure to check the entire paper for consistency. Eg Page 10, Line 12 "Procedural checklists provide step-by-step guidance" does not make sense in the current manuscript with Do-Confirm default, if any procedure were done before EM activation.

AUTHORS RESPOND:

We agree with the reviewer that it does not make sense to use procedural checklists in Do-Confirm mode. We have now written on P11 L13: "Regarding problem and diagnosis checklists, Do-Confirm is the default mode of EM use." We have added the following sentence: "Regarding procedure checklists and fact sheets, the default modes of use are Read-Do and Sampling respectively." (P12 L2)

REVIEWER:

The thoughtful ways that authors have built in the empowering question from EM nurse and the large screen for shared viewing will help activate appropriate use in Do-Confirm mode if there was no EM use yet (or interrupted use). Per study protocol, the documenting nurse will be trained to ask that empowering question at a certain point in patient stability. To clarify, I would suggest still keeping this Do-Confirm mode for standardized EM activation and training of documenting nurse, just not training entire team to a default Do-Confirm mode for the reasons above. Note: How to identify when to activate EM use should be further defined in both methods here and for nurse training protocol, as it is often unclear or forgotten when 'then' is.

AUTHORS RESPOND:

The nurses that are credentialed to manage priority one patients in the RR compose a small group, and these nurses will receive specific training regarding when and how to activate the EM and navigate between checklists. It is logistically impossible to train entire teams, since the physicians managing priority one patients in the RR constitute a large heterogenous group of physicians belonging to several specialties.

According to the protocol, the default mode of use of problem and diagnosis checklists is Do-Confirm. The nurse asks an empowering question once the management plan has been finalized. We agree with the reviewer that it may be ambiguous as to when this juncture has been reached but we do not

believe that this issue can be resolved by adding to the protocol an "Are you done?" type of question, no matter how diplomatically the question is phrased. Rather, we believe that it is best for the documentation nurses to understand the concept of Do-Confirm to be able to ask the empowering question at the correct time. Documentation nurses may specifically ask the physician whether the management plan is finalized. In order to improve the clarity of this concept in the manuscript, we have added: "Once the management plan is finalized, (a juncture that may be verbally confirmed with the team-leader), " on P11 L15 to specify the timing of the empowering question.

Physicians from the department of Emergency Medicine will be informed that the default mode of use is Do-Confirm, i.e. that no checklist will automatically be provided, but that the documentation nurse has been given the task of asking an empowering question to the physician after usual management about which checklist to display, and to read out loud the items on the checklist. Physicians from the department of Emergency Medicine will also be informed that they have the mandate to request checklists/fact sheets at any time, depart from recommendations on the checklists, and ask that the EM not be used.

REVIEWER:

For data analysis, if most leaders/teams do not prompt use separately prior to that, this will achieve the Do-Confirm intended 'default' study design, with patients serving as their own controls as the authors state. And if many leaders/teams choose to initiate use earlier than prompted by documenting nurse, eg actively requesting EM for sampling mode eg of a medication detail or for Read-Do mode in the case of a very unfamiliar event, those cases would show smaller, if any, quantitative changes for the impact of EM on clinically appropriate actions, after EM is used to systematically check actions later.

However, as the authors initially suggested and I agree: the #/% of those cases would make their own important statement about the value clinicians see in the tool, and the other Do-Confirm cases could still be analyzed in the initial ways described.

Question: Will you have or can you think of a way to capture whether the nurse activated EM in Do-Confirm mode (later) by asking an empowering question about which checklist to display? Versus whether a leader/team member requested a specific checklist (earlier; sampling or Read-Do eg procedural checklist or unfamiliar event for those clinicians)?

If so, could consider taking the 50 cases from the former group for a cleaner study. If not, it would seem reasonable still to compare actions before and after EM use, whether that use is activated by documenting nurse or whether that use is requested early by a team member for sampling of key information or Read- Do mode for a procedure etc.

AUTHORS RESPOND:

When problem and diagnosis checklists are used in Do-Confirm mode, the documenting nurse is instructed to read out loud each item and click on one of three buttons featuring on the items row regarding whether the item is

- deemed indicated and has already been carried out
- deemed irrelevant or contraindicated
- deemed indicated and will be carried out thanks to EM use (P11 L19)

By selecting one of three buttons per item on the checklist, the documenting nurse registers that the checklist was used in Do-Confirm mode. When the nurse has not clicked on any of the buttons, the implication is that the checklist was used in Read-Do or Sampling mode.

The purpose of the in-depth analysis of the 50 cases is to find evidence (or lack thereof) that checklist

use improves the quality of care. We have chosen to look exclusively at cases where problem or diagnosis checklists are used as Do-Confirm. Should the physician switch from Do-Confirm to Read-Do before having exhausted his or her ideas regarding investigations and management, it becomes impossible to determine whether subsequently performed investigations and measures were performed thanks to the checklist, or whether these might have been performed anyway without checklist use. We agree that the percentage of cases used as Read-Do or Sampling will provide some information about how clinicians value the tool and integrate it within their practice.

REVIEWER:

b. P11, L7-10 Emergency Manual Protocol for Use

"We recommend Do-Confirm as the default mode of EM use—the team first manages priority 1 patients without support from the EM. The documenting nurse then asks the team: "Which checklist should I display?"

-The bigger comment above suggests that tweaking language here to something like "After team begins patient management with initial actions, the documenting nurse then asks the team: "Which checklist should I display?"

AUTHORS RESPOND:

Should the documenting nurse ask the empowering question "after the team begins patient management with initial actions" and before the juncture where the management plan has been finalized, this would encourage the team to switch into Read-Do mode.

REVIEWER:

-In addition, more detail would be relevant here to define when is 'then' and how will the documenting nurse be triggered to recognize and therefore activate EM use with this empowering question. During an evolving medical crisis, as opposed to a pre-procedure setup, there is no clear moment when 'Do' management is completed and 'Confirm' checking begins. Both briefly for the methods, and then for in more detail for the nurse training, this will be important to define more clearly.

AUTHORS RESPOND:

Regarding the timing of the empowering questions: please see the response provided above. Regarding an evolving crisis: the principle of Do-Confirm still apply even if a new problem (e.g. sudden bradycardia) occurs or a new diagnosis is suspected.

REVIEWER:

It may be helpful to read if not familiar with/cite here if space allows (and/or in subsequent study article) 'implementation intentions' literature for If/Then behavioral triggers, developed originally by Gollwitzer PhD and applied to healthcare:

Classic/Broad References: Gollwitzer P, Sheeran P. Implementation intentions and goal achievement: A metaanalysis of effects and processes. *Adv Exp Soc Psychol.* 2006;38:69–119. or Gollwitzer PM. Implementation intentions: Strong effects of simple plans. *Am Psychol.* 1999;54:493–503. &/or Reference an overview of their application to healthcare, somewhat more recently, eg: Saddawi-Konefka D, Schumacher DJ, Baker KH, Charnin JE, Gollwitzer PM. Changing physician behavior with implementation intentions: closing the gap between intentions and actions. *Academic Medicine.* 2016 Sep 1;91(9):1211-6.

Some ideas to consider for prompting 'If/Then' in the nurses' training:

- If chest compressions have begun but no event has been announced
- If noise level is increasing (chaotic environment)
- If actions are being repeated but patient is not improving
- If get to a certain time point from patient arrival to trauma bay

How you choose If/Then triggers for nurse activation of EM use will be influenced by both ideal EM use, human factors for remembering to trigger use, and this research study design. For the combination of these, given the nurse is also documenting time presumably, it strikes me that the 'certain time point from patient arrival to trauma bay' may be a good way to both trigger nurse activation and to standardize. Authors would know best for deciding in ED RR, what amount of time that would be for the sweetspot goal of not too early to interrupt potentially effective team management and not too late to help with life-saving omissions or course corrections.

AUTHORS RESPOND:

We thank the reviewer for bringing to our attention the concept of implementation intentions! This is a novel concept for us. The core of the concept appears to be that enjoining individuals to formulate concrete "If X then Y" implementation intentions bolsters the likelihood that they will achieve their goals.

It is unclear to us to what extent the concept of implementation intentions applies to the team management of priority one patients in the resuscitation room. In our RR, resuscitation teams carry out a "Sign-In" prior to the arrival of a priority one patient. During the Sign-In, team members share available information about the patient with each other, discuss which conditions the patient may be suffering from, consider which treatments may need to be delivered in the resuscitation room and who does what. This process involves "If X then Y" statements with the goal of shortening reaction time to potential clinical courses. We would argue that the Sign-In is consistent with the CRM injunction to "Fly ahead of the plane": "Anticipation is key for goal-oriented behaviour. Consider the requirements of a case in advance, think of what could be difficult and plan ahead for each possible difficulty. . . . Mentally stay ahead of the game." Rall M, Dieckmann P. Crisis Resource Management to Improve Patient Safety. Refresher Course Lecture ESA 2005.

We agree that the documenting nurse can improve team performance by taking on operative team leadership tasks (while the team-leader physician retains responsibility for strategic leadership). For example, during cardiac arrests, the documentation nurse is responsible for preparing the team for rhythm analysis every second minute and keeping track of when adrenalin is to be administered. During the management of priority one trauma patients, the documenting nurse announces when 10, 15, 20 minutes etc have elapsed since the arrival of the patient.

The documenting nurse will have the added role of asking an empowering question for each priority one patient, activating the EM, displaying checklists and reading aloud through these. We believe that it is in the interest of patient safety to not assign additional roles to, nor in other respects increase the mandate of, the documenting nurse. Once personnel are familiar with the EM and have a sense of how it best can be integrated within clinical care, documenting nurses may be given the mandate to bring forth certain checklists at their own initiative under certain circumstances and enjoin the team to use these.

REVIEWER:

Minor Points:

P5, L7 Typo: "and guiding—though do not dictating—decision-making." Remove the 'do'

AUTHORS RESPOND:

Done. This sentence now features on P5 L7.

REVIEWER:

P7, L22 Typo: 'rational' should be 'rationale' (noun)

AUTHORS RESPOND:

Thank you, corrected. The word now features on P7 L24.

REVIEWER:

P8, L7-16 "Challenges" bullets. Recommend adding (for implementation challenges):

-Activating use

-Enabling shared team use, including a reader, with shared mental model, e.g. via large display

If you wish, it would be relevant to expand slightly on each here, as mentioned above in the novel contribution description.

AUTHORS RESPOND:

We have added the following bullet to the list of challenges: "Ensuring that checklists relevant to the clinical context are made available to all team members at a suitable juncture during patient management, and encouraging checklist use without negatively impacting on team dynamics, are not self-evident tasks." P8 L19

By doing so, we address the challenges of activating use ("made available" + "encouraging checklist use") and enabling shared team use ("available to all team members"). We also include the challenge that the reviewer has previously brought up regarding "when" the EM should be activated.

We mention the documenting nurse's reader function in the section Emergency Manual: Protocol for use as a response to how the mentioned challenge has informed the study design.

We mention the large screen in the section Emergency Manual: content and format as a response to how the mentioned challenge has informed the study design.

We have chosen to not expand on e.g. shared decision-making given that the study does not specifically focus on this/these aspects.

REVIEWER:

This is a challenging to design and an impactful, important study.

AUTHORS RESPOND:

We believe that this study will lay the foundations (in terms of hardware, software, cultural milieu and acquired user-data) for other studies with different designs. We are very grateful for all the reviewer's additions to the manuscript, comments, suggestions, and support for this study!